# Presynaptic Rac1 controls synaptic strength through the regulation of synaptic vesicle priming

Christian Keine[1,2,3], Mohammed Al-Yaari[1], Tamara Radulovic[1,2,3], Connon I Thomas[4], Paula Valino Ramos[1], Debbie Guerrero-Given[4], Mrinalini Ranjan[5,6], Holger Taschenberger[5], Naomi Kamasawa[4], Samuel M Young Jr[1,7]*

[1]Department of Anatomy and Cell Biology, University of Iowa, Iowa City, United States; [2]Department of Human Medicine, Carl-von-Ossietzky University Oldenburg, Oldenburg, Germany; [3]Research Center Neurosensory Science, Carl-von-Ossietzky University Oldenburg, Oldenburg, Germany; [4]Electron Microscopy Core Facility, Max Planck Florida Institute for Neuroscience, Jupiter, United States; [5]Department of Molecular Neurobiology, Max Planck Institute for Multidisciplinary Sciences, Göttingen, Germany; [6]Göttingen Graduate School for Neurosciences, Biophysics, and Molecular Biosciences, Göttingen, Germany; [7]Department of Otolaryngology, Iowa Neuroscience Institute, University of Iowa, Iowa City, United States

**\*For correspondence:**
samuel-m-young@uiowa.edu

**Competing interest:** The authors declare that no competing interests exist.

**Abstract** Synapses contain a limited number of synaptic vesicles (SVs) that are released in response to action potentials (APs). Therefore, sustaining synaptic transmission over a wide range of AP firing rates and timescales depends on SV release and replenishment. Although actin dynamics impact synaptic transmission, how presynaptic regulators of actin signaling cascades control SV release and replenishment remains unresolved. Rac1, a Rho GTPase, regulates actin signaling cascades that control synaptogenesis, neuronal development, and postsynaptic function. However, the presynaptic role of Rac1 in regulating synaptic transmission is unclear. To unravel Rac1's roles in controlling transmitter release, we performed selective presynaptic ablation of Rac1 at the mature mouse calyx of Held synapse. Loss of Rac1 increased synaptic strength, accelerated EPSC recovery after conditioning stimulus trains, and augmented spontaneous SV release with no change in presynaptic morphology or AZ ultrastructure. Analyses with constrained short-term plasticity models revealed faster SV priming kinetics and, depending on model assumptions, elevated SV release probability or higher abundance of tightly docked fusion-competent SVs in Rac1-deficient synapses. We conclude that presynaptic Rac1 is a key regulator of synaptic transmission and plasticity mainly by regulating the dynamics of SV priming and potentially SV release probability.

## Editor's evaluation

Keine et al. study the roles of the RhoGTPase Rac1 and actin in neurotransmitter release by ablating Rac1 at an age when synapses are essentially mature, thereby minimizing developmental compensations. They describe compelling findings supporting an increase in synaptic strength, interpreted as either an increase in release probability or priming of synaptic vesicles. Although direct support for Rac1-dependent altered presynaptic actin is not provided, the study delivers fundamental functional information on the role of Rac1 in regulating presynaptic release.

## Introduction

Information encoding in the nervous system requires synaptic transmission to drive and sustain action potentials (APs) over rapidly changing and highly variable AP firing rates (*Reinagel and Laughlin, 2001*; *Theunissen and Elie, 2014*; *Brette, 2015*; *Azarfar et al., 2018*). However, the number of synaptic vesicles (SVs) available for fusion in response to an AP, the readily releasable pool (RRP), is limited (*Alabi and Tsien, 2012*). Therefore, tight regulation of SV release and RRP replenishment is required for synaptic reliability and temporally precise information encoding (*Neher, 2010*; *Hallermann and Silver, 2013*). Priming is the process that generates fusion-competent SVs. It is a critical step in the SV cycle that primarily regulates RRP size and SV pool replenishment. Priming also controls SV release probability ($P_r$) by determining SV fusogenicity ('molecular priming') and regulating the spatial coupling between docked SV and presynaptic $Ca^{2+}$ entry ('positional priming') (*Klug et al., 2012*; *Schneggenburger and Rosenmund, 2015*; *Neher and Brose, 2018*). In some synapses, the SV priming kinetics are highly dependent on presynaptic cytosolic $Ca^{2+}$ levels, which are activity-dependent. Importantly, human mutations in molecules that regulate priming are associated with neurological disorders (*Waites et al., 2011*; *Wondolowski and Dickman, 2013*; *Torres et al., 2017*; *Bonnycastle et al., 2021*). Therefore, elucidating the molecular mechanisms that regulate SV priming is critical to understanding the diversity of neuronal information encoding in health and disease.

Actin is a central component of both the presynaptic and postsynaptic compartment, with diverse roles in regulating synaptic function and neuronal circuit development. Manipulation of actin dynamics or interference with presynaptic AZ proteins implicated in regulating actin dynamics affects transmitter release and SV replenishment, as well as $P_r$ (*Morales et al., 2000*; *Sakaba and Neher, 2003*; *Cingolani and Goda, 2008*; *Sun and Bamji, 2011*; *Waites et al., 2011*; *Lee et al., 2012*; *Lee et al., 2013*; *Montesinos et al., 2015*; *Rust and Maritzen, 2015*). However, due to disparate results, the role of presynaptic actin signaling cascades in regulating transmitter release and SV pool replenishment is controversial. Finally, in contrast to the postsynaptic compartment, our understanding of how presynaptic regulators of actin signaling cascades control synaptic transmission and short-term plasticity (STP) is in the early stages.

Rac1, a Rho GTPase, is a critical regulator of actin signaling cascades (*Bosco et al., 2009*; *Yasuda, 2017*), and human mutations in Rac1 are associated with neurological disorders (*Bai et al., 2015*; *Reijnders et al., 2017*; *Zamboni et al., 2018*). Rac1 is involved in multiple processes controlling synaptogenesis, axon guidance, neuronal development, and postsynaptic function to regulate neuronal circuit function (*Bai et al., 2015*). Although Rac1 is expressed in both the pre- and postsynaptic compartment (*Threadgill et al., 1997*; *Doussau et al., 2000*; *Kumanogoh et al., 2001*; *O'Neil et al., 2021*), its presynaptic role in regulating synaptic transmission is not well understood. Recent work using cultured hippocampal neurons proposed that presynaptic Rac1 is a negative regulator of SV pool replenishment (*O'Neil et al., 2021*), although at which steps in the SV cycle it exerts its regulatory role is unknown. Finally, how Rac1 regulates the temporal dynamics of transmitter release and pool replenishment in a native neuronal circuit remains elusive.

To unravel the roles of presynaptic Rac1 in regulating transmitter release, we utilized the calyx of Held, a glutamatergic axosomatic presynaptic terminal in the auditory brainstem, which is the sole input to drive AP firing in the principal cells of the medial nucleus of the trapezoid body (MNTB) (*Borst and Soria van Hoeve, 2012*; *Joris and Trussell, 2018*). In the calyx of Held, RRP dynamics are tightly regulated to support a nearly failsafe synaptic transmission with high temporal precision, but the molecular machinery of transmitter release and SV pool replenishment and their regulation are similar to conventional presynaptic terminals in the central nervous system (*Iwasaki and Takahashi, 1998*; *Iwasaki et al., 2000*; *Iwasaki and Takahashi, 2001*; *Sätzler et al., 2002*; *Taschenberger et al., 2002*; *Rollenhagen and Lübke, 2006*; *Neher and Sakaba, 2008*; *Alabi and Tsien, 2012*; *Eggermann et al., 2011*; *Hallermann and Silver, 2013*; *Schneggenburger and Rosenmund, 2015*). In addition, molecular manipulations specific to only the calyceal terminal can be carried out at different developmental stages (*Wimmer et al., 2006*; *Young and Neher, 2009*; *Chen et al., 2013*; *Lübbert et al., 2019*). To elucidate the roles of presynaptic Rac1 in regulating transmitter release while avoiding interference with its role in synaptogenesis, axon guidance, and neuronal development, we selectively deleted Rac1 2 days after hearing onset in postnatal day (P) 14 mouse calyx of Held synapses. At this time point, the synapse is functionally mature, and neuronal properties of brainstem circuits are considered 'adult-like'. Subsequently, we determined how the loss of Rac1 impacted calyx of Held – MNTB

principal cell transmission at the adult stage (P28 onwards) (*Sonntag et al., 2009*; *Crins et al., 2011*; *Sonntag et al., 2011*). Presynaptic Rac1 deletion did neither affect the calyx of Held morphology nor active zone (AZ) ultrastructure but led to increased synaptic strength, faster SV pool replenishment, and augmented spontaneous SV release. Additionally, we found that the loss of Rac1 delayed EPSC onsets and potentiated asynchronous release during high-frequency train stimulation.

Analysis of the experimental data with constrained STP models confirmed faster SV priming kinetics in Rac1-deficient synapses. Methods of quantal analysis, which assume a single and homogenous pool of readily releasable SVs (*Neher, 2015*; *Schneggenburger and Rosenmund, 2015*), reported an increased $P_r$ and a tendency towards an increased RRP after Rac1 loss. Both experimental findings were corroborated in numerical simulation using a single-pool STP model (*Weis et al., 1999*). In contrast, simulations based on a sequential two-step priming scheme which assumes two distinct states of docked/primed SVs, a loosely docked (LS), immature priming state which is not fusion competent, and a tightly docked (TS), mature priming state which is fusion competent (*Neher and Taschenberger, 2021*; *Lin et al., 2022*), required only changes in SV priming kinetics but no change in $P_r$ or the number of release sites to reproduce experimental data. Simulations based on the sequential two-step SV priming and fusion scheme fully accounted for the increased synaptic strength in $Rac1^{-/-}$ synapses by a larger abundance of tightly docked SVs. Therefore, we propose that presynaptic Rac1 is a key molecule that controls synaptic strength and STP primarily by regulating the SV priming dynamics and potentially $P_r$. Finally, we conclude that presynaptic Rac1 is a critical regulator of synaptic transmission and plasticity.

## Results

### Presynaptic deletion of Rac1 after hearing onset does not impact calyx of Held morphology or AZ ultrastructure

Presynaptic terminals contain Rac1 (*Doussau et al., 2000*; *O'Neil et al., 2021*) which regulates synaptogenesis, axon guidance, and neuronal development (*Xu et al., 2019*; *Zhang et al., 2021*). In this study, we aimed to elucidate Rac1's presynaptic function in controlling synaptic transmission and plasticity independent of its role in regulating synapse development and maturation at the calyx of Held synapse. To do so, we injected HdAd co-expressing Cre recombinase and EGFP into the cochlear nucleus (CN) of P14 $Rac1^{flox/flox}$ mice when the calyx of Held synapse is considered 'adult-like' and commenced synapse analysis at P28 onwards (*Figure 1A*; *Sonntag et al., 2009*; *Crins et al., 2011*; *Sonntag et al., 2011*). Since Rac1 controls synapse development and morphology, it was essential to determine whether the loss of Rac1 after hearing onset altered the calyx of Held morphology or AZ ultrastructure. We analyzed calyx morphology from 3D reconstructions of confocal z-stack images acquired from $Rac1^{+/+}$ and $Rac1^{-/-}$ calyces at P28 and found no difference in calyx surface area or volume (*Figure 1B*). To determine if the loss of Rac1 impacted AZ ultrastructure, we performed ultrathin-section TEM and analyzed AZ length, SV distribution, and the number of docked SVs and found no difference between $Rac1^{+/+}$ and $Rac1^{-/-}$ (*Figure 1C*). Therefore, we conclude that, after hearing onset, Rac1 does not regulate calyx of Held morphology or AZ ultrastructure.

### Loss of Rac1 increases synaptic strength and relative synaptic depression

Perturbations of the presynaptic actin cytoskeleton impact synaptic transmission and plasticity in multiple model systems and synapses (*Cole et al., 2000*; *Sakaba and Neher, 2003*; *Bleckert et al., 2012*; *Lee et al., 2012*; *Rust and Maritzen, 2015*; *Miki et al., 2016*; *Gentile et al., 2022*; *Wu and Chan, 2022*). Since Rac1 is an actin cytoskeleton regulator, we examined how the loss of Rac1 impacted AP-evoked synaptic transmission and STP. Afferent fibers of calyx synapses were electrically stimulated with 50 APs at two stimulation frequencies (50 and 500 Hz), representing typical in-vivo firing rates at the calyx. AMPAR-mediated ESPCs were recorded in MNTB principal cells innervated by transduced ($Rac1^{-/-}$) and non-transduced ($Rac1^{+/+}$) calyces in 1.2 mM external $Ca^{2+}$ and at 36–37°C to closely mimic in-vivo conditions (*Figure 2A and B*). By analyzing the initial response ($EPSC_1$) of the EPSC trains, we found a robust increase in synaptic strength upon Rac1 deletion ($Rac^{+/+}$ = 1.3 ± 0.4 nA vs. $Rac1^{-/-}$ = 3 ± 1.1 nA, p < 0.001, n = 15/15, *Figure 2C and D1*) with no change in EPSC waveform. Plotting EPSC amplitudes vs. stimulus number revealed substantial differences in STP between $Rac1^{+/+}$

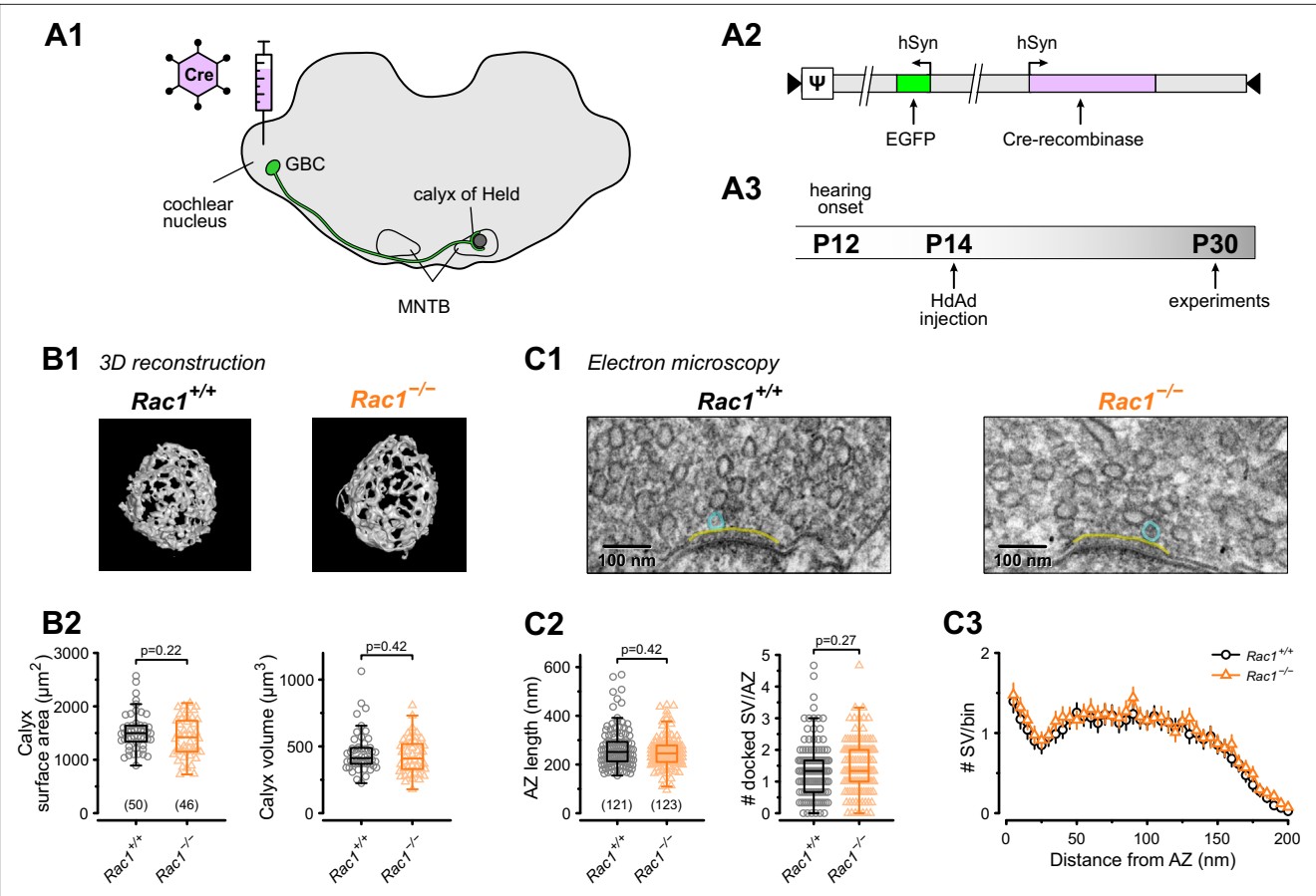

**Figure 1.** Loss of presynaptic Rac1 after hearing onset does not affect calyx of Held gross morphology or ultrastructure. (**A**) Cre recombinase-expressing HdAds were injected into the cochlear nucleus of $Rac1^{fl/fl}$ mice at P14, yielding $Rac1^{-/-}$ calyces of Held. All experiments were performed at around four weeks of age. Cre recombinase-expressing calyces could be visually identified by simultaneous expression of EGFP. (**B1**) Representative reconstruction of calyx terminals of $Rac1^{+/+}$ (left) and $Rac1^{-/-}$ (right) animals. (**B2**) Calyx morphology assessed by surface area (left) and volume (right) was not affected by the loss of Rac1. (**C1**) Representative EM images of the active zone (yellow) and docked SV (blue) to assess synaptic ultrastructure. (**C2**) AZ length and number of docked SV were comparable between $Rac1^{+/+}$ and $Rac1^{-/-}$. (**C3**) SV distribution as a function of distance to AZ was not different between $Rac1^{+/+}$ and $Rac1^{-/-}$. Box plot whiskers extend to the minimum/maximum within the 1.5 interquartile range; open markers indicate individual data points. For EM data, the results of three independent investigators were averaged. All data shown in the figure and the detailed results of statistical tests are part of the supplementary file.

The online version of this article includes the following source data for figure 1:

**Source data 1.** Excel file containing the data shown in *Figure 1* and the results of statistical analysis.

and $Rac1^{-/-}$ synapses. At 50 Hz stimulation, both $Rac1^{+/+}$ and $Rac1^{-/-}$ showed short-term depression, which was more pronounced in $Rac1^{-/-}$ resulting in increased steady-state depression (EPSC$_{ss}$ / EPSC$_1$) (*Figure 2A*). Despite the stronger relative short-term depression, absolute steady-state EPSC amplitudes were almost twofold larger in $Rac1^{-/-}$ ($Rac1^{+/+}$ = 0.47 ± 0.14 nA vs. $Rac1^{-/-}$ = 0.84 ± 0.22 nA, p < 0.001, n = 15/15, *Figure 2D1*). At 500 Hz stimulation, $Rac1^{+/+}$ showed robust short-term facilitation, which was absent in $Rac1^{-/-}$ (Paired-pulse ratio PPR = EPSC$_2$ / EPSC$_1$: $Rac1^{+/+}$ = 1.2 ± 0.1 vs. $Rac1^{-/-}$ = 1 ± 0.1, p < 0.001, n = 15/15, *Figure 2B*). Similar to 50 Hz stimulation, $Rac1^{-/-}$ showed stronger relative steady-state depression at 500 Hz stimulation. Notably, absolute steady-state EPSC amplitudes at 500 Hz stimulation frequency were similar between $Rac1^{+/+}$ and $Rac1^{-/-}$ (*Figure 2D1*).

Since the loss of Rac1 increased synaptic strength and altered STP, we aimed to identify the underlying mechanisms and evaluated RRP size and SV release probability ($P_r$) using established quantal analysis methods (*Elmqvist and Quastel, 1965*; *Neher, 2015*; *Thanawala and Regehr, 2016*). We estimated RRP size using 500 Hz stimulus trains which effectively depleted the RRP in both $Rac1^{+/+}$ and $Rac1^{-/-}$ synapses by applying three conventional paradigms based on the common assumption of quantal

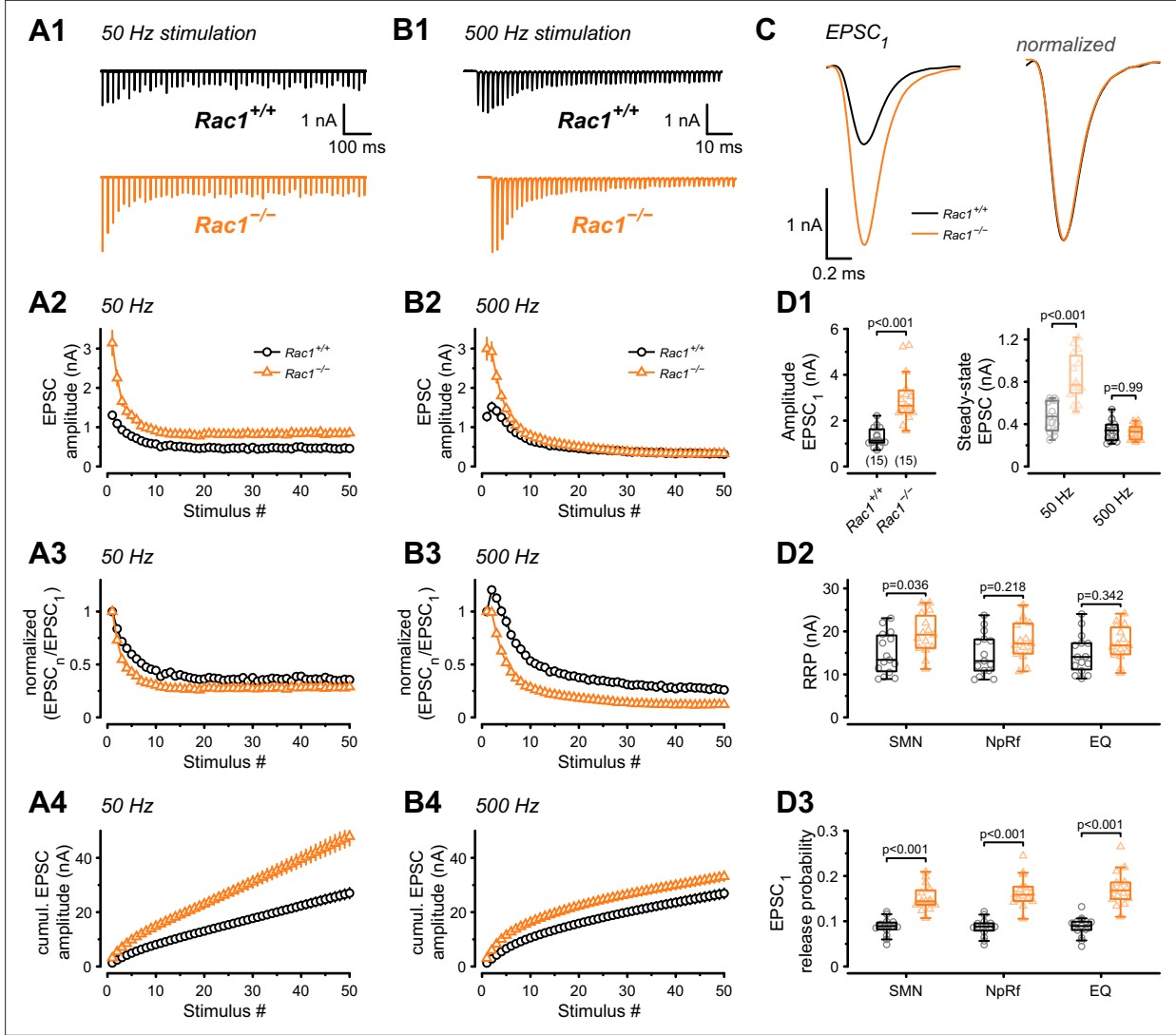

**Figure 2.** Presynaptic Rac1 regulates synaptic vesicles release probability and synaptic strength. Synaptic transmission at the calyx of Held – MNTB synapse was studied using different stimulation frequencies at P28 after deletion of Rac1 at P14. (**A1, B1**) Representative evoked EPSCs for *Rac1*[+/+] (black) and *Rac1*[−/−] (orange) at 50 Hz and 500 Hz stimulation frequency. Stimulus artifacts were blanked for clarity. (**C**) Magnification of the first EPSC (EPSC$_1$). Ablation of presynaptic Rac1 resulted in increased EPSC$_1$ amplitude with no change in EPSC dynamics. (**A2–A4**) At 50 Hz stimulation frequency, *Rac1*[−/−] showed stronger short-term depression despite larger steady-state EPSC amplitudes. (**B2–B4**) At 500 Hz stimulation frequency, loss of Rac1 resulted in a lack of short-term facilitation and increased synaptic depression with no change in steady-state EPSC amplitude. (**D1**) Population data showing an increase in EPSC$_1$ amplitude in *Rac1*[−/−]. Steady-state EPSC amplitudes were increased in *Rac1*[−/−] at 50 Hz but not at 500 Hz stimulation frequency. (**D2**) Population data of the readily releasable pool (RRP) using three different estimation methods, suggesting little to no change in RRP size (**D3**) Population data indicating that EPSC$_1$ release probability in *Rac1*[−/−] was elevated independent of estimation method. All data shown in the figure and the detailed results of statistical tests are part of the supplementary file.

The online version of this article includes the following source data for figure 2:

**Source data 1.** Excel file containing the data shown in *Figure 2* and the results of statistical analysis.

release originating from a single and functionally homogenous pool of readily-releasable SVs (*Neher, 2015*; *Schneggenburger and Rosenmund, 2015*): EQ, NpRf, and SMN with correction (*Elmqvist and Quastel, 1965*; *Neher, 2015*; *Thanawala and Regehr, 2016*). All three methods reported a moderate increase in RRP size in *Rac1*[−/−] calyces (*Figure 2D2*), however, this was statistically significant only for the SMN analysis. The initial $P_r$ of resting synapses was estimated by dividing the EPSC$_1$ amplitude by the estimated RRP sizes. All three analysis methods revealed an approximately twofold increase in $P_r$ (SMN: *Rac1*[+/+] = 0.09 ± 0.02 vs. *Rac1*[−/−] = 0.15 ± 0.03, p < 0.001; NpRf: *Rac1*[+/+] = 0.09 ± 0.02 vs. *Rac1*[−/−] = 0.17 ± 0.03, p < 0.001; EQ: *Rac1*[+/+] = 0.09 ± 0.02 vs. *Rac1*[−/−] = 0.17 ± 0.04, p < 0.001,

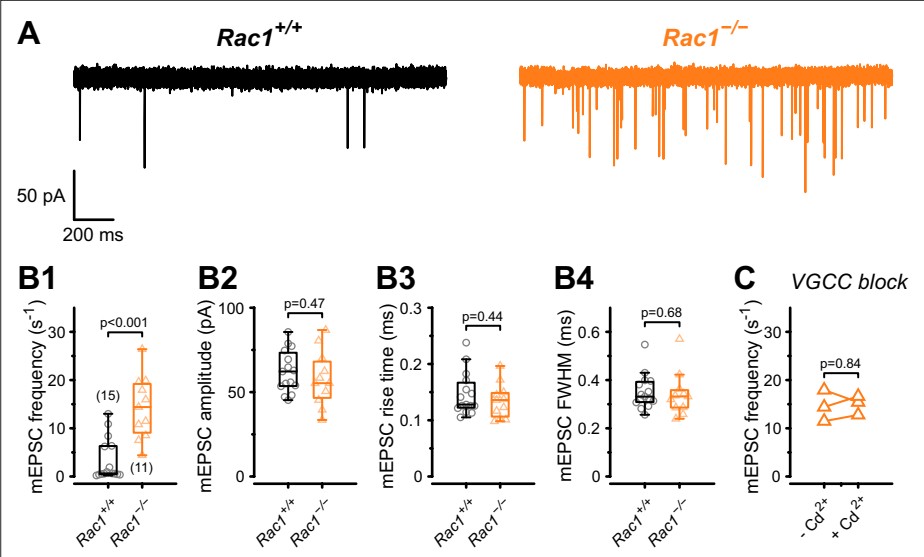

**Figure 3.** Presynaptic loss of Rac1 increases calcium-independent neurotransmitter release. (**A**) Representative recordings of mEPSCs for *Rac1*$^{+/+}$ (left, black) and *Rac1*$^{-/-}$ (right, orange). (**B1–B4**) Rac1 deletion increased mEPSC frequency but did not affect mEPSC amplitude, rise time, or full width at half-maximal amplitude (FWHM). (**C**) The increased mEPSC rates at *Rac1*$^{-/-}$ were independent of presynaptic voltage-gated calcium channels (VGCC), as blocking VGCC with cadmium (Cd$^{2+}$) did not affect mEPSC frequency. All data shown in the figure and the detailed results of statistical tests are part of the supplementary file.

The online version of this article includes the following source data for figure 3:

**Source data 1.** Excel file containing the data shown in *Figure 3* and the results of statistical analysis.

n = 15/15, *Figure 2D3*). Therefore, based on the assumption of a single and functionally homogenous RRP, this analysis indicates that presynaptic Rac1 deletion increases synaptic strength and short-term depression primarily by elevating initial $P_r$ with little increase in RRP size. This suggests that Rac1 controls synaptic strength as a negative regulator of $P_r$.

## Loss of Rac1 increases mEPSC frequency but not amplitude

The RRP size and the $P_r$ of fusion-competent SVs is determined by the SV priming process, which involves the assembly of the molecular fusion apparatus, defined as 'molecular priming'. In addition, $P_r$ also depends on the spatial coupling between docked SVs and Ca$^{2+}$ entry sites which may be adjusted by a distinct 'positional priming' step. Thus, 'molecular priming' encompasses the steps that render SVs fusion competent and regulate their intrinsic fusogenicity (*Basu et al., 2007*; *Xue et al., 2010*; *Schneggenburger and Rosenmund, 2015*; *Schotten et al., 2015*), while positional priming consists of the steps that place molecularly primed SVs close to voltage-gated calcium channels (VGCCs) (*Neher, 2010*). The spatial coupling between SV and VGCCs critically determines the speed and efficacy of AP-evoked release (*Eggermann et al., 2011*; *Stanley, 2016*). Spontaneous SV release is not or only little dependent on VGCCs (*Schneggenburger and Rosenmund, 2015*; *Kavalali, 2020*), thus the frequency of miniature EPSC (mEPSCs) can be interpreted as a readout of intrinsic SV fusogenicity at basal Ca$^{2+}$ with increased SV fusogenicity causing higher mEPSC frequencies (*Basu et al., 2007*; *Schotten et al., 2015*; *Dong et al., 2018*). Therefore, to determine if an increased intrinsic SV fusogenicity caused the increase in $P_r$, we recorded mEPSCs from *Rac1*$^{+/+}$ and *Rac1*$^{-/-}$ calyx synapses (*Figure 3*) and found a four-fold increase in mEPSC frequency (*Rac1*$^{+/+}$ = 3.4 ± 4.4 Hz vs. *Rac1*$^{-/-}$ = 14.3 ± 6.5 Hz, p < 0.001, n = 15/11) with no change in mEPSC amplitude or waveform. To rule out that the increased mEPSC frequencies were due to changes in presynaptic Ca$^{2+}$ currents, we recorded mEPSCs in the presence of 200 µM Cd$^{2+}$, a non-specific VGCC blocker. Since Cd$^{2+}$ did not affect mEPSC frequencies we conclude that Rac1 loss increases intrinsic SV fusogenicity at basal Ca$^{2+}$.

## Loss of Rac1 increases EPSC onset delays and decreases synchronicity of AP-evoked release

Although we found an increase in SV fusogenicity, this does not rule out an additional role for Rac1 in regulating spatial coupling distances between molecularly primed SVs and VGCCs. In the mature calyx of Held, AP-evoked SV release is governed by local $Ca^{2+}$ nanodomains, ensuring a fast onset and highly synchronous AP-evoked EPSCs to faithfully encode auditory information (*Fedchyshyn and Wang, 2005*). In addition to the gating kinetics of presynaptic VGCCs and postsynaptic AMPARs,

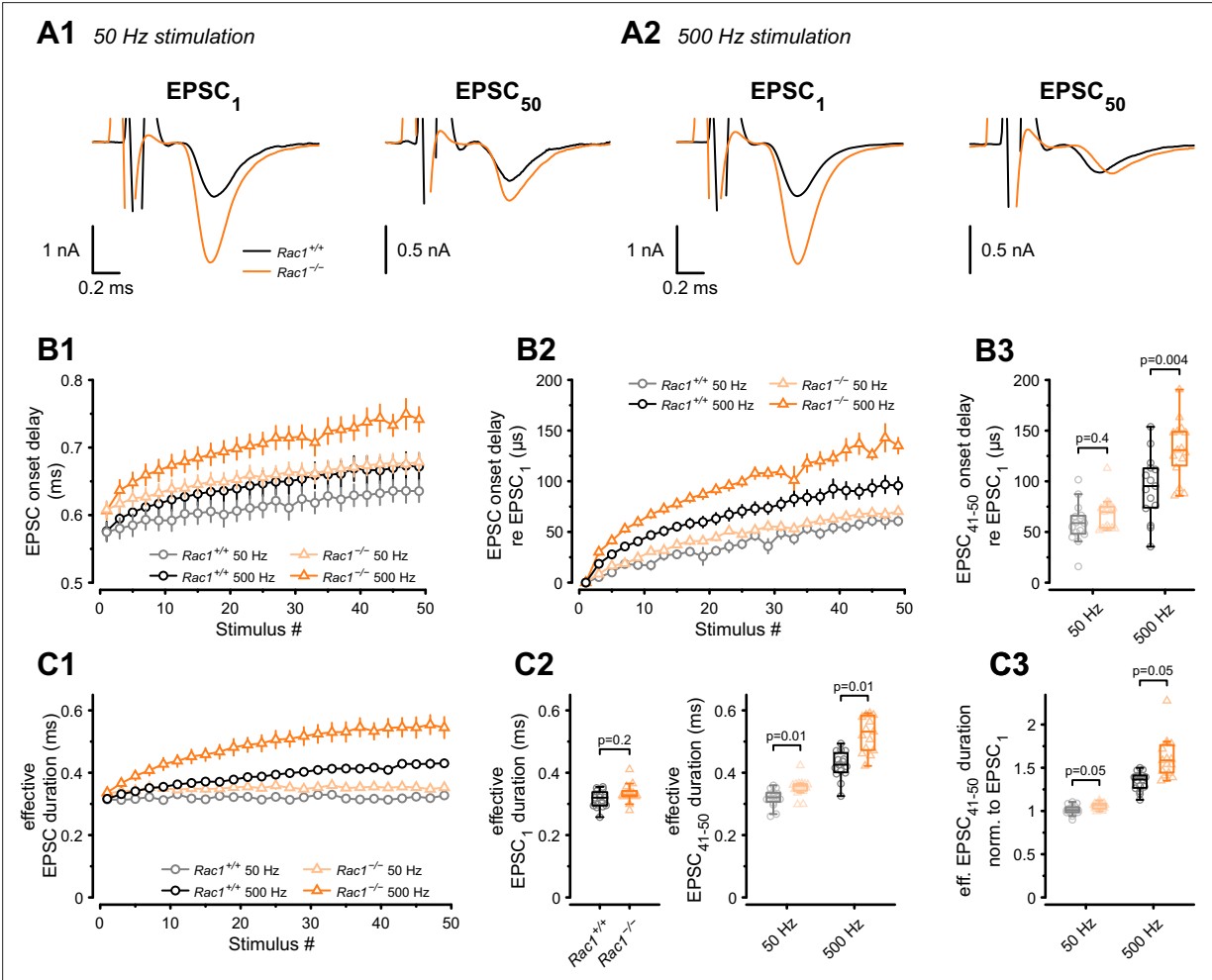

**Figure 4.** Presynaptic loss of Rac1 decreases SV synchronicity and prolongs EPSC onset at high-frequency stimulation. (**A**) Experiments were performed at low (50 Hz, **A1**) and high (500 Hz, **A2**) stimulation frequencies. Representative recordings of first ($EPSC_1$) and last ($EPSC_{50}$) EPSC in the stimulus train. Traces are aligned at the EPSC onset of $EPSC_1$. Stimulus artifacts are partially blanked for better visibility. Note the shift in the onset of $EPSC_{50}$ in $Rac1^{-/-}$ compared to $Rac1^{+/+}$ at 500 Hz but not 50 Hz. (**B1**) Absolute EPSC onset delay for 50 Hz (gray and light orange) and 500 Hz (black and orange) stimulation. (**B2**) EPSC onset delay relative to $EPSC_1$ for 50 Hz (gray and light orange) and 500 Hz (black and orange). At 50 Hz, the EPSC onset delay was similar between $Rac1^{+/+}$ and $Rac1^{-/-}$. At 500 Hz, the EPSC onset delay was substantially larger at $Rac1^{-/-}$. For better visualization, only every second data point is shown. (**B3**) EPSC onset delay of the last ten EPSCs relative to $EPSC_1$ for 50 Hz and 500 Hz stimulation. EPSC delay of the last ten EPSCs in the stimulus train ($EPSC_{41-50}$) was not different between $Rac1^{+/+}$ and $Rac1^{-/-}$ at 50 Hz but increased for $Rac1^{-/-}$ at 500 Hz stimulation frequency. (**C1**) Analysis of 'effective EPSC duration' to estimate SV release synchronicity during 50 Hz and 500 Hz stimulation. Synchronicity was estimated from 'effective EPSC duration' by dividing the EPSC charge by the EPSC amplitude. Note the increase in effective EPSC duration for $Rac1^{-/-}$ at 500 Hz stimulation. For better visualization, only every second data point is shown (**C2**) EPSC duration was not different for $EPSC_1$ but slightly longer for late EPSCs in $Rac1^{-/-}$ at 50 Hz and substantially longer at 500 Hz stimulation frequency. (**C3**) Effective EPSC duration of $EPSC_{41-50}$ normalized to $EPSC_1$. Note the progressive increase in effective EPSC duration in $Rac1^{-/-}$ with increasing stimulation frequency. All data shown in the figure and the detailed results of statistical tests are part of the supplementary file.

The online version of this article includes the following source data for figure 4:

**Source data 1.** Excel file containing the data shown in *Figure 4* and the results of statistical analysis.

the time between presynaptic AP and EPSC onset (EPSC onset delay) is determined by the coupling distance between SVs and VGCCs. The coupling distance defines the time for $Ca^{2+}$ to diffuse and bind to the $Ca^{2+}$ sensor and initiate SV fusion (*Fedchyshyn and Wang, 2007*; *Nakamura et al., 2015*). Thus, EPSC onset delays can serve as a readout of changes in spatial coupling distances, as increased onset delays are consistent with SVs being more loosely coupled to VGCCs and vice versa. Therefore, we measured EPSC onset delays during 50 Hz and 500 Hz stimulation (*Figure 4A and B*) and found them to become progressively larger during stimulation for *Rac1*$^{+/+}$ and *Rac1*$^{-/-}$ calyces. At 50 Hz, the increase in EPSC onset delays during stimulus trains was comparable between *Rac1*$^{+/+}$ and *Rac1*$^{-/-}$, amounting to about 60 µs between the first and the last 10 EPSCs. At 500 Hz stimulation, however, EPSC onset delays increased more rapidly in *Rac1*$^{-/-}$, with steady-state EPSC onset delays being significantly larger for *Rac1*$^{-/-}$ compared to *Rac1*$^{+/+}$ (*Rac1*$^{+/+}$ = 94 ± 32 µs vs. *Rac1*$^{-/-}$ = 131 ± 29 µs, p = 0.004, n = 15/15).

In addition to modulating EPSC onset delays, coupling distances between SV and VGCCs affect the time course of synchronous release and the relative contribution of synchronous vs. asynchronous release during AP trains (*Wadel et al., 2007*; *Chen et al., 2015*; *Stanley, 2016*; *Yang et al., 2021*). This is because synchronous release is dominated by tightly coupled SVs which rapidly fuse in response to high local $[Ca^{2+}]$, while asynchronous release likely represents a stronger contribution of more loosely coupled SVs (*Sakaba, 2006*; *Schneggenburger and Rosenmund, 2015*). An approximate measure for changes in the time course of AP-evoked release can be obtained by analyzing the EPSC charge over EPSC amplitude ratio ('effective EPSC duration') representing the width of a square current pulse with the same amplitude as the EPSC peak and same integral as the EPSC charge. Therefore, we calculated the effective EPSC duration for both 50 Hz and 500 Hz stimulation (*Figure 4C*) and found its value for EPSC$_1$ comparable between *Rac1*$^{+/+}$ and *Rac1*$^{-/-}$. At steady-state during 50 Hz stimulation, however, the effective EPSC duration was slightly longer in *Rac1*$^{-/-}$ (*Rac1*$^{+/+}$ = 0.32 ± 0.03 ms vs. *Rac1*$^{-/-}$ = 0.35 ± 0.03 ms, p = 0.006). At steady-state during 500 Hz stimulation, the effective EPSC duration in *Rac1*$^{-/-}$ calyces was prolonged further and increased by ~25% compared to *Rac1*$^{+/+}$ (*Rac1*$^{+/+}$ = 0.43 ± 0.04 ms vs. *Rac1*$^{-/-}$ = 0.55 ± 0.12 ms, p < 0.001). These findings are consistent with transmitter release being less synchronous in *Rac1*$^{-/-}$ synapses, especially during sustained activity at high AP firing rates.

In summary, we found that *Rac1*$^{-/-}$ synapses had longer EPSC onset delays and showed more strongly increasing effective EPSC durations during stimulus trains, especially at high stimulation frequencies, implying less synchronous release. Since tighter SV to VGCCs coupling has the opposite effect, that is, generates shorter EPSC onset delays and more tightly synchronized release, we conclude that the increase of synaptic strength in *Rac1*$^{-/-}$ calyces is not due to tighter spatial coupling between SVs and VGCCs.

## Loss of Rac1 facilitates EPSC recovery and RRP replenishment

The kinetics of molecular priming regulates RRP replenishment and determines steady-state release rates during high-frequency stimulation (*Lipstein et al., 2013*; *Ritzau-Jost et al., 2018*; *Lipstein et al., 2021*). Since *Rac1* deletion increased steady-state release during 50 Hz stimulus trains, we hypothesized that SV pool replenishment proceeds faster in the absence of Rac1. To test how Rac1 loss influences RRP replenishment, we applied afferent fiber stimulation using a paired train protocol consisting of a 500 Hz conditioning train (50 APs) followed by a second 500 Hz test train at varying recovery intervals (*Figure 5*). Recovery was then measured for both the initial EPSC amplitude (EPSC$_{test}$) and the RRP estimate of the test trains. Recovery of the initial EPSC amplitude was quantified in terms of both its absolute (*Figure 5A2*) and its fractional value (*Figure 5A3*), with the latter being the ratio (EPSC$_{test}$ − EPSC$_{ss}$) / (EPSC$_1$ − EPSC$_{ss}$), where EPSC$_{test}$, EPSC$_1$ and EPSC$_{ss}$ are the initial amplitude of the test train, and the first and the steady-state amplitudes of the 500 Hz conditioning train, respectively. Recovery of absolute EPSC$_{test}$ amplitude was significantly different between *Rac1*$^{+/+}$ and *Rac1*$^{-/-}$ (*Rac1*$^{+/+}$: A = 1.37, $\tau_{fast}$ = 28 ms, $\tau_{slow}$ = 2.7 s, $f_{slow}$ = 0.86, $\tau_w$ = 2.3 s vs. *Rac1*$^{-/-}$: A = 3.4, $\tau_{fast}$ = 44 ms, $\tau_{slow}$ = 2.3 s, $f_{slow}$ = 0.83, $\tau_w$ = 1.9 s, p < 0.001, n = 15/15) and fractional EPSC recovery was over 50% faster in *Rac1*$^{-/-}$ (*Rac1*$^{+/+}$: $\tau_{fast}$ = 40 ms, $\tau_{slow}$ = 2.8 s, $f_{slow}$ = 0.88, $\tau_w$ = 2.4 s vs. *Rac1*$^{-/-}$: $\tau_{fast}$ = 47 ms, $\tau_{slow}$ = 2 s, $f_{slow}$ = 0.76, $\tau_w$ = 1.5 s, p < 0.001, *Figure 5A*). Next, we analyzed fractional RRP recovery by dividing the RRP estimate of the test train by the RRP estimate of the conditioning train and found that RRP recovery rates were about 40% faster in *Rac1*$^{-/-}$ (*Rac1*$^{+/+}$:

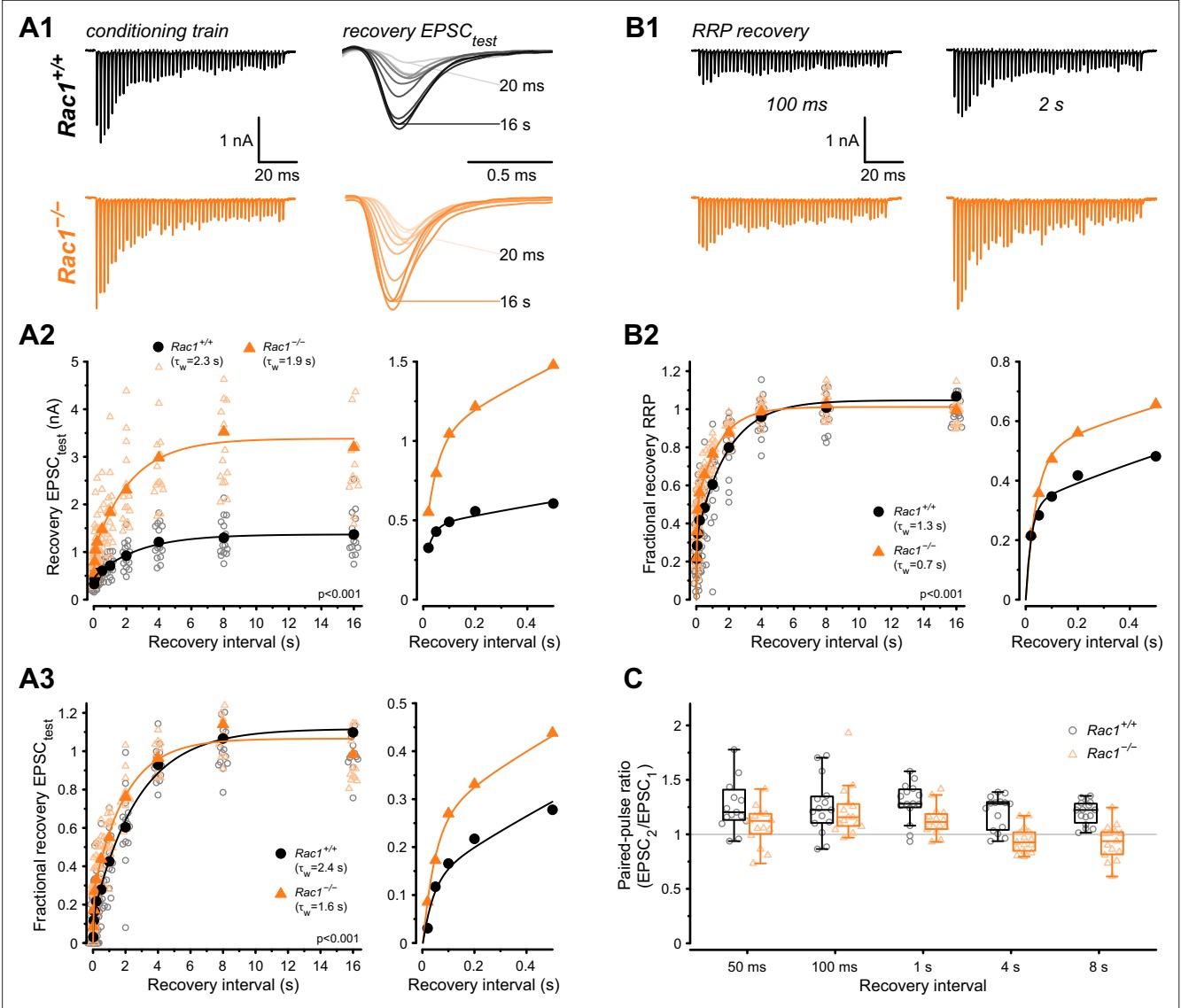

**Figure 5.** Loss of presynaptic Rac1 facilitates synaptic vesicle recovery. Recovery of single EPSC (EPSC$_{test}$) and RRP recovery was measured by two consecutive train stimuli (conditioning stimulus and recovery stimulus) at 500 Hz at varying recovery intervals. (**A**) Single EPSC recovery. (**A1**) Representative traces for *Rac1$^{+/+}$* (black) and *Rac1$^{-/-}$* (orange) for recovery intervals ranging from 20 ms to 16 s. (**A2**) Recovery of absolute EPSC amplitudes as a function of recovery interval with a magnification of short intervals (right). (**A3**) Fractional EPSC recovery as a function of recovery interval. (**B**) Recovery of the RRP. (**B1**) Representative recovery traces following a 100 ms and 2 s recovery interval. The conditioning stimulus train is the same as in A1. (**B2**) Fractional RRP recovery was faster in *Rac1$^{-/-}$* compared to *Rac1$^{+/+}$*. (**C**) Recovery of the paired-pulse ratio (PPR) of the first two EPSCs of the recovery train. PPR was consistently lower in *Rac1$^{-/-}$* but the difference was emphasized at longer recovery intervals. All data shown in the figure and the detailed results of statistical tests are part of the supplementary file.

The online version of this article includes the following source data for figure 5:

**Source data 1.** Excel file containing the data shown in *Figure 5* and the results of statistical analysis.

$\tau_{fast}$ = 22 ms, $\tau_{slow}$ = 1.9 s, $f_{slow}$ = 0.7, $\tau_w$ = 1.3 s vs. *Rac1$^{-/-}$*: $\tau_{fast}$ = 40 ms, $\tau_{slow}$ = 1.4 s, $f_{slow}$ = 0.52, $\tau_w$ = 0.7 s, p < 0.001, *Figure 5B*). Finally, we compared the PPR (EPSC$_2$ / EPSC$_1$) of the test train after different recovery intervals. Independent of recovery interval, PPR was consistently lower in *Rac1$^{-/-}$* (*Rac1$^{+/+}$* = 1.3 ± 0.1 vs. *Rac1$^{-/-}$* = 1.1 ± 0.1, p < 0.001, *Figure 5C*), consistent with an increase in $P_r$.

## Numerical simulations of STP and EPSC recovery are consistent with altered SV priming after Rac1 loss

Since the loss of presynaptic Rac1 caused three principal changes in synaptic function: (*i*) increased synaptic strength (*Figure 2D1*), (*ii*) increased steady-state release during 50 Hz stimulation (*Figure 2A2*), and (*iii*) accelerated EPSC recovery following conditioning 500 Hz trains (*Figure 5*), we next sought to corroborate our conclusions about underlying synaptic mechanisms by reproducing experimental data in numerical simulations. To do so, we used two distinct STP models: (1) a single-pool model with a $Ca^{2+}$-dependent SV pool replenishment similar to the release-site model of *Weis et al., 1999* and (2) a recently established sequential two-step priming scheme (*Neher and Taschenberger, 2021*; *Lin et al., 2022*).

The single-pool model assumes a single type of release site to which SVs can reversibly dock, and SV docking and priming is described by a single transition step (*Figure 6A1*). The kinetics of the forward (priming) transition (determined by the rate constant $k_f$) is characterized by a Michaelis-Menten-like dependence on cytosolic $[Ca^{2+}]$ (*Figure 6A2* inset) while the backward (unpriming) transition has a fixed rate constant ($k_b$). For resting synapses, the equilibrium between empty sites (ES) and sites occupied by a docked and primed SV (DS) is given by the ratio $k_f/k_b$ at basal $[Ca^{2+}]$ in this scheme. The total number of SV docking sites ($N_{total}$), priming and unpriming rate constants ($k_f$ and $k_b$), Ca-dependence of the priming step (parameters $\sigma$ and $K_{0.5}$), release probability $P_r$, and the time course of $[Ca^{2+}]$ regulating the priming speed ('effective $[Ca^{2+}]$') were free parameters and adjusted by trial and error (*Figure 6—figure supplement 1*) to reproduce experimentally observed differences in the time course of fractional recovery (*Figure 6A3*), initial synaptic strength, and time course of STP during 50 and 500 Hz stimulation (*Figure 6A4*). A comparison between simulated and experimental data shows that the single-pool model can describe the functional differences between $Rac1^{+/+}$ and $Rac1^{-/-}$ synapses, reproducing larger initial synaptic strength, larger steady-state release during 50 Hz trains in Rac1-deficient synapses, and similar steady-state release during 500 Hz trains in both genotypes (*Figure 6A5*).

To describe both $Rac1^{+/+}$ and $Rac1^{-/-}$ synapses adequately, we had to introduce changes in $N_{total}$, priming kinetics and $P_r$. The best fit was achieved by adjusting the ratio $k_f/k_b$ at basal $[Ca^{2+}]$ to yield a number of release sites occupied with a docked/primed SV at rest (RRP) of 2150 and 2532 and by setting $P_r$ to 0.08 and 0.165 for $Rac1^{+/+}$ and $Rac1^{-/-}$ synapses, respectively (*Figure 6—figure supplement 1*). Larger initial synaptic strength in $Rac1^{-/-}$ synapses resulted from ~1.2-fold and ~twofold changes in RRP and $P_r$, respectively, which is consistent with the analysis shown in *Figure 2*, which rests on similar assumptions as the single-pool STP model. The higher steady-state release in $Rac1^{-/-}$ synapses is primarily a result of their higher priming rate constant $k_f$ for effective $[Ca^{2+}]$ up to ~1 μM, while a faster saturation of $k_f$ with effective $[Ca^{2+}]$ above ~1 μM in $Rac1^{-/-}$ synapses (*Figure 6A2*) accounts for steady-state release during 500 Hz stimulation being similar to $Rac1^{+/+}$ (*Figure 6A5*).

Subsequently, we simulated the experimental data using a two-step priming scheme, which postulates a sequential build-up of the SV fusion apparatus and distinguishes two distinct priming states: an immature loosely docked state (LS) and a mature tightly-docked (TS) state (*Neher and Brose, 2018*; *Neher and Taschenberger, 2021*; *Lin et al., 2022*). The two-step priming scheme (*Figure 6B1*) reproduces functional changes in $Rac1^{-/-}$ synapses, including the accelerated EPSC recovery after pool depletion (*Figure 6B3*), increased initial EPSC amplitudes, and elevated steady-state release during 50 Hz trains (*Figure 6B4*). In contrast to simulations using the single-pool model (*Figure 6A1*), the sequential two-step model required adjustments only to the model parameters determining the SV priming kinetics to reproduce the changes observed in $Rac1^{-/-}$ synapses (*Figure 6B1*). In resting $Rac1^{-/-}$ synapses, the priming equilibrium was shifted towards a higher fraction of TS SVs in addition to a slight reduction of the fraction of empty sites (ES) because of the increased priming rate constants $k_1$ and $k_2$ at resting cytosolic $[Ca^{2+}]$. The higher abundance of fusion-competent SVs in resting $Rac1^{-/-}$ synapses ($Rac1^{+/+}$ = 691 TS SVs vs. $Rac1^{-/-}$ = 1666 TS SVs) fully accounts for their increased synaptic strength, while the $P_r$ and the total number of sites $N_{total}$ were constrained to the same values for $Rac1^{+/+}$ and $Rac1^{-/-}$ synapses. Proper adjustment of the $Ca^{2+}$-dependence of $k_1$ and $k_2$ (*Figure 6B2*) reproduces the different steady-state release in $Rac1^{-/-}$ compared to $Rac1^{+/+}$ synapses (*Figure 6B4*). The steeper increase in $k_1$ and $k_2$ with increasing $[Ca^{2+}]$ (*Figure 6B2*) accounts for the faster recovery early after cessation of 500 Hz conditioning when $[Ca^{2+}]$ is still elevated (*Figure 6B3*). Thus, the sequential two-step model is capable of replicating the observed changes in $Rac1^{-/-}$ synapses (*Figure 6B3*

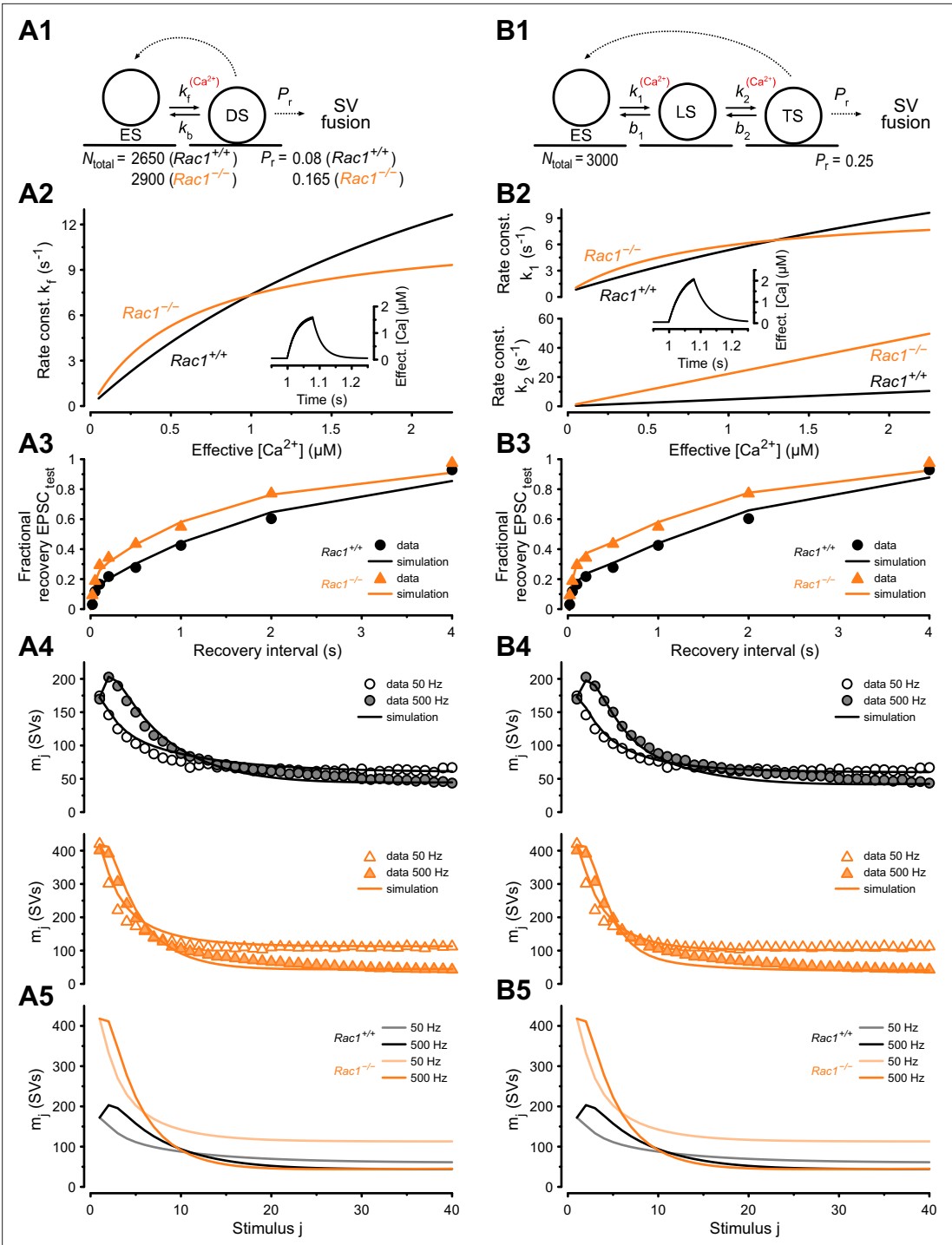

**Figure 6.** Numerical simulations of 50 and 500 Hz STP and EPSC recovery after conditioning 500 Hz trains are consistent with Rac1-loss induced changes in SV priming. Experimental observations were equally well reproduced by either of two kinetic schemes of SV priming and fusion: a single-pool model (**A**) or a recently proposed (**Lin et al., 2022**) sequential two-step SV priming scheme (**B**). (**A1**) Diagram of vesicle states for the single-pool model. SVs reversibly dock at empty release sites (ES). SVs in the docked and primed state (DS) undergo fusion with the probability $P_r$ upon AP arrival. Vacated sites become immediately available for SV docking and priming. Transitions represented by dashed lines occur instantaneously, while those represented by solid lines occur with rate constants as indicated. Forward transition rate constant is $Ca^{2+}$-dependent. For the single-pool model, $P_r$ and total number of sites ($N_{total}$) were free parameters for both genotypes. The model predicts an increased $P_r$ from 0.08 ($Rac1^{+/+}$) to 0.165 ($Rac1^{-/-}$) and an increase in the number of docked SVs (RRP) from 2150 SVs ($Rac1^{+/+}$) to 2532 SVs ($Rac1^{-/-}$). (**A2**) Dependence of $k_f$ on cytosolic $[Ca^{2+}]$ (effective $[Ca^{2+}]$) for $Rac1^{+/+}$ (black) and $Rac1^{-/-}$ (orange) synapses. The inset illustrates the time course of the effective $[Ca^{2+}]$ during a 500 Hz train consisting of 40 stimuli. (**A3**) Predictions of the single-pool model (lines) for the time course of the fractional recovery of $EPSC_{test}$ after 500 Hz conditioning trains superimposed

*Figure 6 continued on next page*

*Figure 6 continued*

onto experimental data for $Rac1^{+/+}$ (black circles) and $Rac1^{-/-}$ (orange triangles) synapses (data from *Figure 5A3*). (**A4**) Predictions of the single-pool model (lines) for the time course of STP during 50 Hz and 500 Hz trains superimposed onto experimental data (circles) for $Rac1^{+/+}$ (black, top panel) and $Rac1^{-/-}$ (orange, bottom panel) synapses. (**A5**) Model predictions for the time course of STP during 50 Hz and 500 Hz trains for $Rac1^{+/+}$ (gray and black) and $Rac1^{-/-}$ (light and dark orange) synapses shown superimposed to facilitate comparison. (**B1**) Diagram of vesicle states for the sequential two-step priming scheme. SVs reversibly dock at empty release sites (ES) and become fusion-competent by undergoing a sequence of two priming steps. After initial docking, SVs in the loosely docked state (LS) reversibly transition to the tightly docked state (TS) from which they undergo fusion upon AP arrival with the probability $P_r$. Vacated sites become immediately available for SV docking and priming. Transitions represented by dashed lines occur instantaneously, while those represented by solid lines occur with rate constants as indicated. Forward transition rate constants are $Ca^{2+}$-dependent. For the sequential two-step priming scheme, $P_r$ and total number of sites ($N_{total}$) were constrained to the same values for $Rac1^{+/+}$ and $Rac1^{-/-}$ synapses and only parameters determining the kinetics of the two priming steps were allowed to differ between genotypes. (**B2**) Dependence of $k_1$ (top panel) and $k_2$ (bottom panel) on cytosolic $[Ca^{2+}]$ for $Rac1^{+/+}$ (black) and $Rac1^{-/-}$ (orange) synapses. The inset illustrates the time course of the effective $[Ca^{2+}]$ during a 500 Hz train consisting of 40 stimuli. (**B3**) Predictions of the sequential two-step model (lines) for the time course of the fractional recovery of $EPSC_{test}$ after 500 Hz conditioning trains superimposed onto experimental data for $Rac1^{+/+}$ (black circles) and $Rac1^{-/-}$ (orange triangles) synapses (data from *Figure 5A3*). (**B4**) Predictions of the sequential two-step model (lines) for the time course of STP during 50 Hz and 500 Hz trains superimposed onto experimental data (circles) for $Rac1^{+/+}$ (black, top panel) and $Rac1^{-/-}$ (orange, bottom panel) synapses. (**B5**) Model predictions for the time course of STP during 50 Hz and 500 Hz trains for $Rac1^{+/+}$ (gray and black) and $Rac1^{-/-}$ (light and dark orange) synapses shown superimposed to facilitate comparison.

The online version of this article includes the following figure supplement(s) for figure 6:

**Figure supplement 1.** Model predictions and parameters of simple single-pool models fitted to $Rac1^{+/+}$ and $Rac1^{-/-}$ STP and recovery data sets.

**Figure supplement 2.** Model predictions and parameters of sequential two-step models fitted to $Rac1^{+/+}$ and $Rac1^{-/-}$ STP and recovery data sets.

*and B5*) solely by modifying the priming equilibrium between LS and TS SVs in resting synapses and carefully adjusting the $Ca^{2+}$-dependence of the two SV priming steps with unaltered model parameter values for $P_r$ and $N_{total}$.

In summary, the experimental data available do not allow us to unambiguously favor one model over the other (*Figure 6A1 vs. B1*). Both models reproduce differences in STP and EPSC recovery between $Rac1^{-/-}$ and $Rac1^{+/+}$ synapses, and both models require increased SV priming speed at resting and intermediate $[Ca^{2+}]$ in $Rac1^{-/-}$ synapses to reproduce the data faithfully. While the single-pool model (*Figure 6A1*) requires different values for $N_{total}$ and $P_r$, to account for the changes observed between $Rac1^{-/-}$ and $Rac1^{+/+}$ synapses, necessary changes to model parameters were limited to those determining the SV priming equilibrium ($k_1$, $b_1$, $k_2$, $b_2$) when using the sequential two-step model (*Figure 6B1*).

## Loss of Rac1 impacts delay and temporal precision of AP firing in response to SAM stimuli

Animal vocalization, including human speech, is characterized by rapid amplitude modulations (*Joris et al., 2004*). These amplitude modulations can be mimicked by sinusoidal amplitude-modulated stimuli (SAM), which will produce periodic fluctuations in the firing rates of the calyx of Held (*Mc Laughlin et al., 2008*; *Tolnai et al., 2008*). Therefore, different SAM frequencies will differentially stress SV release and recovery mechanisms and can be used to reveal how the observed changes in synaptic strength and SV pool recovery potentially impact auditory signaling in $Rac1^{-/-}$ calyces. Using in vivo responses to SAM stimuli (*Tolnai et al., 2008*) which contained modulation frequencies between 20 and 2000 Hz as templates for afferent fiber stimulation, we recorded postsynaptic APs in the loose-patch configuration (*Figure 7*). First, we analyzed the success probability of presynaptic stimuli triggering a postsynaptic AP to assess the reliability of synaptic transmission and found no difference between $Rac1^{+/+}$ and $Rac1^{-/-}$ for all modulation frequencies tested (*Figure 7B1*). Since temporal precision is crucial in auditory signaling, we then analyzed if loss of Rac1 affected the temporal precision of postsynaptic AP generation by calculating the standard deviation of AP delays ('AP jitter'). For all modulation frequencies, temporal precision was unchanged when analyzed for the complete stimulus (*Figure 7B2*). However, our previous analysis showed that EPSC onset delay and synchronicity were only affected at high stimulation frequencies. Since the SAM stimuli generate periods of high firing activity interspersed with periods of low activity, we tested how preceding activity influenced postsynaptic AP jitter and AP delay. First, we used the preceding inter-spike interval (ISI) to estimate preceding activity and calculated AP jitter and AP delay for different ISIs (*Figure 7C1*). Both AP jitter and AP delay were comparable between $Rac1^{+/+}$ and $Rac1^{-/-}$ for most ISIs,

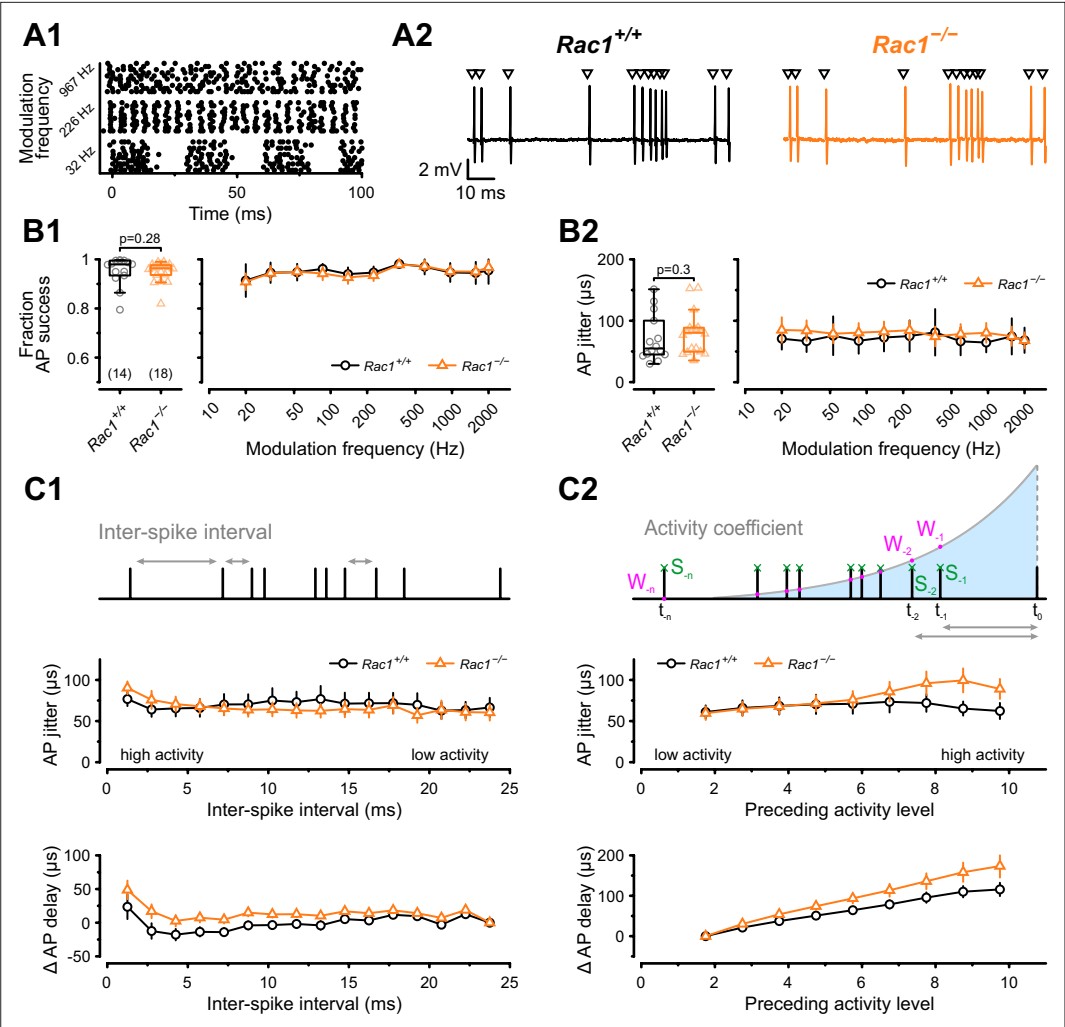

**Figure 7.** Alterations in presynaptic release probability did not impair the reliability of postsynaptic action potential generation but decreased temporal precision during in vivo-like activity. (**A1**) AP firing was recorded in response to in vivo-like stimulation patterns derived from responses to sinusoidal amplitude-modulated sounds at different modulation frequencies. Raster plot shows three representative stimulation patterns. (**A2**) Representative traces of loose-patch recordings during afferent fiber stimulation with in vivo-like activity for *Rac1*$^{+/+}$ (black) and *Rac1*$^{-/-}$ (orange). Triangles indicate the stimulus time points. Stimulus artifacts were blanked for clarity. (**B1**) Fraction of successful AP generation was not different between *Rac1*$^{+/+}$ and *Rac1*$^{-/-}$ independent of modulation frequency (**B2**) AP jitter defined as the standard deviation of AP latencies was not changed in *Rac1*$^{-/-}$ neither for the complete stimulus nor as a function of amplitude modulation frequency. (**C1**) AP jitter and AP delay were largely independent of preceding inter-spike interval. (**C2**) AP jitter and AP delay as a function of preceding activity level. The preceding activity was calculated as the sum of all preceding APs (green, S$_{-1}$, S$_{-2}$, …, S$_{-n}$) weighted by their temporal distance to the AP under observation (purple, W$_{-1}$, W$_{-2}$, …, W$_{-n}$). The weighting was implemented as an exponentially decaying kernel (blue shaded area). Note the increased AP jitter and AP delay in *Rac1*$^{-/-}$ at higher activity levels. All data shown in the figure and the detailed results of statistical tests are part of the supplementary file.

The online version of this article includes the following source data for figure 7:

**Source data 1.** Excel file containing the data shown in *Figure 7* and the results of statistical analysis.

but AP delay in *Rac1*$^{-/-}$ increased slightly at short ISIs. Previous studies showed that EPSC/EPSP amplitudes, and subsequent AP generation, are influenced by preceding neuronal activity (*Haustein et al., 2008*; *Englitz et al., 2009*; *Lorteije et al., 2009*; *Sonntag et al., 2011*; *Yang and Xu-Friedman, 2015*; *Ghanbari et al., 2020*). Since the analysis shown in *Figure 7C1* only considered the last ISI, it may result in an incomplete estimate of preceding activity. Therefore, we additionally estimated

activity levels by calculating the sum of all preceding APs weighted by their distance to the AP under observation (*Figure 7C2*). Using this approach, we found that AP jitter was similar in $Rac1^{+/+}$ and $Rac1^{-/-}$ synapses for low to moderate activity levels but diverged at increasing activity levels, with $Rac1^{-/-}$ showing higher AP jitter ($Rac1^{+/+}$ = 65 ± 32 µs vs. $Rac1^{-/-}$ = 99 ± 62 µs, p < 0.001, n = 14/18). Likewise, AP delays were similar for low activity levels but increased more strongly in $Rac1^{-/-}$ with increasing activity ($Rac1^{+/+}$ = 110 ± 52 µs vs. $Rac1^{-/-}$ = 158 ± 101 µs, p = 0.03). These data suggest that the activity-dependent increase in EPSC onset and reduction in EPSC synchronicity in $Rac1^{-/-}$ at high stimulation frequencies translates into a small but consistent increase in postsynaptic AP delay and AP jitter, thereby potentially decreasing the temporal fidelity of signal transmission.

## Discussion

By genetically ablating Rac1 at the calyx of Held after hearing onset, we identified presynaptic roles of Rac1 in regulating synaptic transmission and plasticity in a native neuronal circuit. Based on our experimental data and numerical simulations, we identify presynaptic Rac1 as a key regulator of synaptic strength and SV pool replenishment by controlling SV priming kinetics and, depending on model assumptions, by either regulating $P_r$ (single-pool model) or by regulating the abundance of tightly docked and fusion-competent SVs (sequential two-step priming model). In conclusion, we propose that presynaptic Rac1 is a critical regulator for encoding information flow in neuronal circuits.

### Presynaptic Rac1 regulates initial synaptic strength

Our finding that Rac1 regulates initial synaptic strength contrasts with a recent study in primary cultured hippocampal neurons which concluded that presynaptic Rac1 had no such effect (*O'Neil et al., 2021*). Multiple reasons may explain these differences. One cause may be due to the specific conditions of a native neuronal circuit which are not fully replicated under in vitro conditions in cultured neuronal circuits. We specifically abated Rac1 at an advanced developmental stage, two days after hearing onset, at which neuronal circuit properties are well defined, and the calyx of Held is considered functionally mature (*Englitz et al., 2009*; *Sonntag et al., 2009*; *Sonntag et al., 2011*; *Borst and Soria van Hoeve, 2012*). Although Rac1 was ablated at 10 days in vitro in the study by *O'Neil et al., 2021*, the corresponding developmental in-vivo stage is difficult to estimate. In addition, it is unknown how and to what extent culture conditions determine synaptic transmission and STP characteristics of hippocampal synapses developing in vitro. Another critical difference between our study and that of *O'Neil et al., 2021* was the recording condition. We used 1.2 mM external [$Ca^{2+}$] and physiological temperature to mimic in vivo conditions (*Lorteije et al., 2009*; *Borst, 2010*; *Forsberg et al., 2019*), while *O'Neil et al., 2021* performed experiments at 2 mM external [$Ca^{2+}$] and room temperature. As a result, our estimates for initial $P_r$ based on the ratio of $EPSC_1$ over RRP had a mean of ~0.1 in $Rac1^{+/+}$ calyces, and values for individual synapses never exceeded 0.15, which allowed us to observe a two-fold increase following Rac1 ablation. In the cultured neurons, using a similar analysis, a mean initial $P_r$ of ~0.4 was reported for $Rac1^{+/+}$ with values for individual synapses frequently exceeding 0.5 in both excitatory and inhibitory neurons (*O'Neil et al., 2021*). It is therefore possible that in cultured neurons, $P_r$ was close to saturation, thereby occluding any further increase in initial synaptic strength in $Rac1^{-/-}$ neurons. In addition, many regulatory steps in the SV cycle are temperature-dependent (*Chanaday and Kavalali, 2020*), and in mouse hippocampal synapses, actin-dependent synaptic release enhancement is restricted to physiological temperatures (*Jensen et al., 2007*).

### Presynaptic Rac1 regulates SV pool replenishment

The availability of fusion-competent SVs critically determines synaptic strength during ongoing stimulation. The steady-state occupancy of the RRP varies with stimulation rates (*Hallermann and Silver, 2013*; *Neher, 2015*) and is determined by the SV pool replenishment kinetics, which are critical for maintaining synaptic transmission. Pool replenishment is typically quantified by two methods (*Hallermann and Silver, 2013*; *Neher, 2015*): (1) the replenishment rate during ongoing stimulation can be obtained as the slope of line fits to the steady-state portion of cumulative EPSC trains assuming that during ongoing stimulation quantal release is balanced by newly replenished SVs. (2) Replenishment rate constants after stimulation can be estimated from the time constants of exponential fits to the

time course of the fractional EPSC or pool recovery plotted as a function of the recovery interval following conditioning trains.

We found that $Rac1^{-/-}$ calyces had larger steady-state EPSC amplitudes, and the slope of the line fits to cumulative EPSC trains revealed a faster replenishment rate at 50 Hz stimulation, similar to observations in hippocampal cultures at 20 and 40 Hz stimulation (*O'Neil et al., 2021*). In contrast, during ongoing 500 Hz stimulation, steady-state EPSC amplitudes and slope of the line fits were similar in $Rac1^{+/+}$ and $Rac1^{-/-}$ synapses. However, time courses of EPSC or pool recovery after 500 Hz conditioning showed an almost 50% faster recovery when fit by bi-exponential functions. This acceleration was largely due to a speed-up of $\tau_{slow}$ which, according to our simulations, reflects a faster recovery occurring at cytosolic $[Ca^{2+}]$ relatively close to resting values. In addition, we estimated a slightly larger fraction of the fast recovery component in $Rac1^{-/-}$ synapses, which, according to our simulations, reflects the magnitude of $Ca^{2+}$-dependent acceleration of SV pool replenishment immediately after cessation of 500 Hz conditioning while cytosolic $[Ca^{2+}]$ decays back to resting values. These observations are consistent with the experimental finding of an enhanced steady-state release in $Rac1^{-/-}$ synapses during 50 Hz stimulation, which reflects their faster pool replenishment at $[Ca^{2+}]<1$ μM because of the steeper increase of the forward priming rate constants with increasing cytosolic $[Ca^{2+}]$. The steady-state release was similar during 500 Hz trains when the synapses regenerate fusion-competent SVs at similar maximum rates in $Rac1^{+/+}$ and $Rac1^{-/-}$. This was primarily caused by a stronger saturation of the Ca-dependence of the priming rate constant for $[Ca^{2+}]>1$ μM in $Rac1^{-/-}$ synapses. Because of the similar steady-state EPSC levels measured at 500 Hz, it is unlikely that Rac1 controls an upstream limit on the priming process.

## Presynaptic Rac1 regulation of spontaneous release

At the mature calyx of Held synapse, blocking presynaptic $Ca^{2+}$ influx through VGCCs does not change spontaneous release rates (*Dong et al., 2018*). Thus, mEPSC frequencies can be interpreted as a readout of intrinsic SV fusogenicity at basal $[Ca^{2+}]$ with higher mEPSC frequencies reflecting increased SV fusogenicity (*Basu et al., 2007*; *Schotten et al., 2015*; *Dong et al., 2018*), provided that pool size remains unaltered. However, SVs undergoing AP-evoked and spontaneous release may not necessarily originate from the same SV pool. Nevertheless, all SVs have to undergo a priming step to acquire fusion competence. Actin is found throughout the presynaptic terminal and application of latrunculin causes a rapid increase in mEPSCs rates in the presence of internal $Ca^{2+}$ chelators (BAPTA-AM, EGTA-AM) while also augmenting AP-evoked release (*Morales et al., 2000*). Thus, although Rac1 loss may differentially affect mEPSCs rates and rates of AP-evoked release, increased mEPSC frequencies following Rac1 loss are consistent with a higher fusogenicity of those SVs contributing to spontaneous release presumably due to Rac1's role as actin regulator.

## Presynaptic Rac1 does not affect the spatial coupling between docked SVs and VGCCs

After docking to release sites, SVs become fusion-competent during the assembly of the molecular fusion machinery, which defines the intrinsic SV fusogenicity. In addition to the intrinsic SV fusogenicity, the probability of SVs undergoing fusion upon AP arrival is determined by their spatial coupling distances to VGCCs, which determines the local $[Ca^{2+}]$ 'seen' by the vesicular $Ca^{2+}$ sensor for fusion (*Neher, 2010*; *Schneggenburger and Rosenmund, 2015*). Therefore, the proximity of SVs to VGCCs (also termed 'positional priming') is a critical determinant of transmitter release (*Wadel et al., 2007*; *Chen et al., 2015*; *Stanley, 2016*). At the calyx of Held after hearing onset, AP-evoked SV release is controlled by local $[Ca^{2+}]$ nanodomains generated around $Ca_V2.1$ channels which are located at ~25 nm distance from docked SVs (*Fedchyshyn and Wang, 2005*; *Chen et al., 2015*; *Nakamura et al., 2015*), resulting in fast synchronous transmitter release. Since Rac1 regulates actin dynamics, loss of Rac1 may result in tighter spatial coupling between SV and VGCCs, thereby increasing synaptic strength. If SVs were located in closer proximity to VGCCs, we would expect to see a shorter EPSC onset delays during both 50 and 500 Hz stimulation. However, we found that EPSC onset delays in $Rac1^{-/-}$ calyx synapses were longer and EPSCs less synchronous, particularly at high firing rates, which is inconsistent with SVs being more tightly coupled to VGCCs.

The EPSC onset delay includes the AP conduction delay, the time between triggering an AP by afferent fiber stimulation and its arrival at the calyx terminal, and the transmitter release delay, the

time between presynaptic AP and SV fusion which is dependent on the distance between SVs to VGCCs (*Fedchyshyn and Wang, 2007*). Therefore, we cannot exclude that loss of Rac1 may prolong conduction delay, thereby obscuring a role of Rac1 in regulating SV to VGCC coupling distances. However, EPSC onset delays of the first EPSC or during 50 Hz train stimuli were similar between $Rac1^{+/+}$ and $Rac1^{-/-}$ calyces. The EPSC onset delay was only increased for steady-state EPSCs during 500 Hz stimulation. In addition, if longer conduction delays were solely responsible for the increase in EPSC onset delays, no changes in release kinetics would be expected. However, during both 50 and 500 Hz stimulus trains, the effective EPSC duration increased more strongly in $Rac1^{-/-}$ calyces compared to $Rac1^{+/+}$ consistent with a less synchronized release time course in the former, possibly because of a larger fraction of SVs located distally from VGCCs contributing to release in $Rac1^{-/-}$.

Another possibility is that presynaptic APs in $Rac1^{-/-}$ were broader and thereby increased presynaptic $Ca^{2+}$ influx which contributed to increased synaptic strength and faster EPSC recovery. Although broader presynaptic APs are expected to widen the release transient and may delay presynaptic $Ca^{2+}$ influx occurring mainly during AP repolarization (*Borst and Sakmann, 1998*; *Li et al., 2007*; *Wang et al., 2008*; *Kochubey et al., 2009*) and thereby increase synaptic delays, we did not observe an increased effective $EPSC_1$ duration or longer $EPSC_1$ onset delays in $Rac1^{-/-}$ calyces. Even though we cannot exclude a Rac1-dependent regulation of presynaptic AP waveform, we do not consider such scenario very likely. Using TEA, a Kv3 channel blocker, to broaden the presynaptic AP impairs high-frequency firing at the calyx of Held (*Wang et al., 1998*; *Wang and Kaczmarek, 1998*; *Johnston et al., 2010*). Since $Rac1^{-/-}$ calyces were able to follow high-frequency stimulation and EPSC onset delays were similar for $EPSC_1$ and throughout the 50 Hz stimulus train it is unlikely that loss of Rac1 caused a general broadening of APs. Therefore, in the absence of experimental evidence supporting Rac1 loss-induced changes in SV to VGCC coupling distances, we propose that Rac1 regulates synaptic strength and RRP recovery at the level of molecular priming, either by increasing the intrinsic fusogenicity of SVs and their $P_r$ or by increasing the abundance of fusion-competent tightly docked SVs. In both scenarios, increased $P_r$ (single-pool model) or a shift in priming equilibrium in favor of TS SVs (two-step model) could account for the increased initial synaptic strength in $Rac1^{-/-}$ synapses and, at least in part, for their higher mEPSC rates.

Although our experimental data do not allow us to favor one scenario over the other unambiguously, both models suggest that Rac1 regulates SV priming. While it was not required to postulate differences in $P_r$ when simulating $Rac1^{+/+}$ and $Rac1^{-/-}$ synapse with a sequential two-step model, we cannot rule out that $P_r$ is indeed changed after Rac1 loss. It is well possible to implement a higher $P_r$ in $Rac1^{-/-}$ calyces within the framework of a sequential two-step model and future experiments combined with simulations will be needed to explore this possibility further.

Loss of Rac1 might also impact endocytosis through changes in actin signaling (*Wu and Chan, 2022*). However, at the calyx of Held, the endocytic role of F-actin appears to be negligible as actin depolymerization using latrunculin did not affect endocytosis (*Eguchi et al., 2017*; *Piriya Ananda Babu et al., 2020*). Furthermore, endocytosis acts on a time scale of seconds and would be too slow to affect SV pool replenishment during short stimulus trains (*Armbruster and Ryan, 2011*). Although ultrafast endocytosis occurs on a timescale of tens of milliseconds it is unlikely to contribute to the $Rac1^{-/-}$ phenotype as SV reformation is too slow with a time scale of tens of seconds (*Watanabe et al., 2013a*; *Watanabe et al., 2013b*). In addition, endocytosis is postulated to play a role in release site clearance (*Neher, 2010*; *Sakaba et al., 2013*) as the perturbation of endocytosis increases the rate of synaptic depression and slows RRP recovery (*Wen et al., 2013*) which we did not observe in the $Rac1^{-/-}$ calyces.

## Rac1 regulation of auditory signaling and information processing

The ability to accurately encode sound information requires synaptic transmission in the lower auditory brainstem to drive and sustain precise AP firing over rapid and large fluctuations of AP firing rates up to the kilohertz range (*Grothe et al., 2010*; *Borst and Soria van Hoeve, 2012*; *Friauf et al., 2015*). The calyx of Held – MNTB principal cell synapse is a failsafe and reliable synaptic relay that transforms afferent AP spike patterns from the CN into precisely timed inhibition to several mono- and binaural nuclei (*Friauf and Ostwald, 1988*; *Spirou et al., 1990*; *Joris et al., 2004*; *Englitz et al., 2009*; *Lorteije et al., 2009*; *Sonntag et al., 2011*). The use of SAM stimuli which mimics the neuronal response to environmental sounds (*Joris et al., 2004*; *Tolnai et al., 2008*), suggests that loss of

Rac1 did not deteriorate the faithful auditory signaling properties over different amplitude modulation frequencies. However, we found that increased EPSC onset delays and decreased SV release synchronicity in *Rac1⁻ᐟ⁻* translated into larger AP delay and AP jitter at high activity levels. Although the absolute changes were modest, the need for high temporal precision in the auditory brainstem might cause a more severe impact in downstream nuclei, such as the lateral and medial superior olive. Since the calyx of Held generates suprathreshold EPSPs even under short-term depression (*Lorteije et al., 2009*; *Lorteije and Borst, 2011*), changes in synaptic strength caused by Rac1 ablation may affect postsynaptic AP timing but should have little effect on postsynaptic AP success probability in MNTB principal neurons. Synaptic transmission at other auditory synapses in the SPON, MSO, or LSO that operate close to AP threshold or rely on synaptic integration to encode information is likely more substantially affected by Rac1 ablation.

## Ideas and speculation

### Rac1 regulation of F-actin dynamics controls SV priming

F-actin regulates several SV cycle steps, which could impact SV replenishment rates (*Morales et al., 2000*; *Sakaba and Neher, 2003*; *Cingolani and Goda, 2008*; *Sun and Bamji, 2011*; *Waites et al., 2011*; *Lee et al., 2012*; *Lee et al., 2013*; *Montesinos et al., 2015*; *Rust and Maritzen, 2015*). Specifically, F-actin may act as a physical barrier within the presynaptic AZ (*Cingolani and Goda, 2008*) as the application of latrunculin transiently increases $P_r$ and mEPSC rates (*Morales et al., 2000*). Studies in which Septin 5 (*Yang et al., 2010*) or the actin cytoskeleton was disrupted (*Lee et al., 2012*) support the idea of F-actin as a structural barrier potentially impacting molecular priming or SV-to-VGCC coupling (*Yang et al., 2010*). Assuming that SV docking distance relative to the plasma membrane corresponds to SV priming (*Imig et al., 2014*; *Jung et al., 2016*; *Pulido and Marty, 2018*), we propose that Rac1's regulation of F-actin may impact the physical barrier and thus affect the conversion from LS to TS, although the $P_r$ of the final TS state in Rac1-deficient terminals remains unknown. Based on our simulations, we speculate that the loss of Rac1 results in local actin depolymerization in the AZ, thereby promoting the LS to TS transition and/or increasing $P_r$ of fusion-competent SVs by allowing vesicle docking closer to the plasma membrane (*Figure 8*). In support of our model, increased synaptic strength has been demonstrated to correlate with shortened tethering filaments that resulted in SVs positioned more closely to the presynaptic membrane (*Jung et al., 2021*). It is also in line with the finding that GIT proteins (*Montesinos et al., 2015*), Arp2/3 (*O'Neil et al., 2021*), and Piccolo (*Waites et al., 2011*), which are all regulators of F-actin dynamics, also control synaptic strength and suggests that molecules regulating F-actin dynamics at the AZ could be key regulators of SV replenishment.

Mutations in Rac1 that result in loss or gain of function are associated with intellectual disability (*Reijnders et al., 2017*; *Zamboni et al., 2018*). While much attention has focused on Rac1 dysregulation in the dendritic compartment or the role of Rac1 in neuronal development, our work demonstrates that presynaptic loss of Rac1 increases synaptic strength and EPSC recovery independent

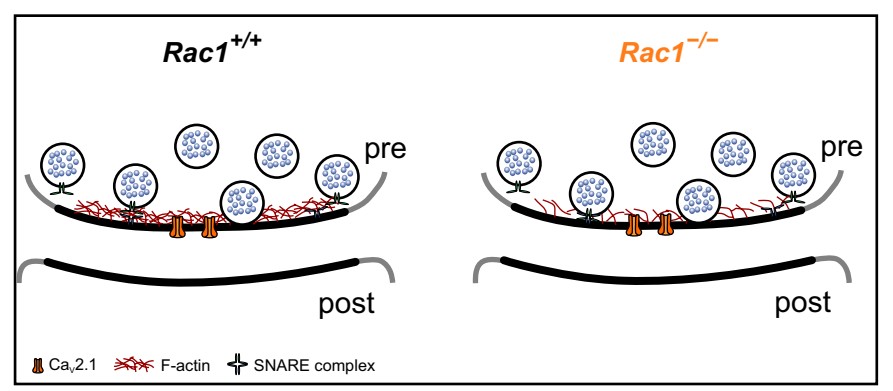

**Figure 8.** Proposed model of Rac1's presynaptic role in regulating synaptic transmission. In the proposed model, loss of Rac1 results in changes in F-actin at the active zone, thereby reducing the physical barrier between SVs and the plasma membrane which increases synaptic strength through faster SV priming and potentially higher $P_r$.

of its developmental role. Human mutations of Rac1 may affect synaptic strength at many different synaptic connections and potentially alter the excitation-inhibition balance and synaptic information processing in neuronal circuits associated with neurological disorders and addiction (*Dietz et al., 2012*; *Bai et al., 2015*; *Wright et al., 2020*).

# Methods

**Key resources table**

| Reagent type (species) or resource | Designation | Source or reference | Identifiers | Additional information |
|---|---|---|---|---|
| Strain, strain background (*Mus musculus*) | Rac1$^{tm1Djk}$/J (Rac1$^{flox/flox}$) | Jackson Laboratory; *Glogauer et al., 2003* | RRID:IMSR_JAX:005550 | either sex |
| Antibody | Anti-GFP (rabbit polyclonal) | Abcam | Cat# ab6556 RRID:AB_305564 | EM (0.1 µg/mL) |
| Antibody | 6 nm colloidal Gold-AffiniPure anti-rabbit IgG (donkey polyclonal) | Jackson ImmunoResearch | Cat# 711-195-152 RRID:AB_2340609 | EM (1:100) |
| Sequence-based reagent | primer: 5'-TCC AAT CTG TGC TGC CCA TC-3' | *Glogauer et al., 2003* | | |
| Sequence-based reagent | primer: 5'-GAT GCT TCT AGG GGT GAG CC-3' | *Glogauer et al., 2003* | | |
| Recombinant DNA reagent | HdAd 28E4 hsyn iCre EGFP (viral vector) | Samuel M. Young, Jr., University of Iowa | | |
| Recombinant DNA reagent | HdAd 28E4 hsyn iCre mEGFP (viral vector) | Samuel M. Young, Jr., University of Iowa | | |
| Recombinant DNA reagent | HdAd 28E4 hsyn mEGFP (viral vector) | Samuel M. Young, Jr., University of Iowa | | |
| Chemical compound, drug | Kynurenic acid | Tocris Bioscience | Cat# 0223 | |
| Chemical compound, drug | Lidocaine N-ethyl bromide (QX-314) | Sigma-Aldrich | Cat# L5783 | |
| Chemical compound, drug | D-AP5 | Tocris Bioscience | Cat# 0106 | |
| Chemical compound, drug | (-)-bicuculline methochloride | Tocris Bioscience | Cat# 0131 | |
| Chemical compound, drug | Strychnine hydrochloride | Tocris Bioscience | Cat# 2785 | |
| Chemical compound, drug | Tetraethylammonium chloride | Sigma-Aldrich | Cat# T-2265 | |
| Chemical compound, drug | Tetrodotoxin | Alomone labs | Cat# T-550 | |
| Chemical compound, drug | Cadmium chloride hemi(pentahydrate) | Sigma-Aldrich | Cat# C3141 | |
| Software, algorithm | Matlab | The Mathworks | RRID:SCR_001622; v9.10 | |
| Software, algorithm | Patchmaster | HEKA; Harvard Bioscience | RRID:SCR_000034; v2x90.2 | |
| Software, algorithm | Igor Pro | Wavemetrics | RRID:SCR_000325; v6.37 | |
| Software, algorithm | Fiji | https://fiji.sc/ | RRID:SCR_002285 | |
| Software, algorithm | Patcher's Power Tools | Max Planck Institute for Biophysical Chemistry; Gottingen; Germany | RRID:SCR_001950; v2.19 | |
| Software, algorithm | StereoDrive | Neurostar | N/A; v3.1.5 | |
| Software, algorithm | Live Acquisition | Thermo Fisher Scientific | N/A; v2.1.0.10 | |
| Software, algorithm | Neuromatic | *Rothman and Silver, 2018* | RRID:SCR_004186 | |

All experiments were performed following animal welfare laws and approved by the Institutional Committee for Care and Use of Animals at the University of Iowa PHS Assurance No. D16-00009 (A3021-01) (Animal Protocol 0021952) and complied with accepted ethical best practices. Animals were housed at a 12 hr light/dark cycle and had access to food and water ad libitum. Experiments were performed on Rac1$^{tm1Djk}$ ($Rac1^{fl/fl}$) mice ($Glogauer\ et\ al.,\ 2003$; RRID:IMSR_JAX:005550, The Jackson Laboratory, Bar Harbor, USA) of either sex. Animals of this line possess a $loxP$ site flanking exon 1 of the $Rac1$ gene disrupting Rac1 expression after recombination mediated by Cre recombinase. Genotyping was performed using PCR-amplification with the following primers 5'-TCC AAT CTG TGC TGC CCA TC-3' and 5'-GAT GCT TCT AGG GGT GAG CC-3' and amplification products ($Rac1^{fl/fl}$: 242 bp, wildtype: 115 bp) were separated by gel electrophoresis on a 1.5% agarose gel ($Glogauer\ et\ al.,\ 2003$). Viral vectors were injected at postnatal day 14 (P14) and experiments were performed at P28-30. All available measures were taken to minimize animal pain and suffering.

## Animals

### DNA construct and recombinant viral vector production

Helper-dependent adenoviral vectors (HdAd) expressing a codon-optimized Cre recombinase (Cre) ($Shimshek\ et\ al.,\ 2002$) were produced as previously described ($Montesinos\ et\ al.,\ 2016$; $Lübbert\ et\ al.,\ 2017$). These HdAd vectors contain two transgene cassettes that independently express Cre and either EGFP or myristoylated EGFP (mEGFP) under the control of the 470 bp human synapsin promoter ($Lübbert\ et\ al.,\ 2017$). In brief, the expression cassette with Cre was cloned into the pdelta28E4 synEGFP plasmid using the $AscI$ enzyme digestion site. The final plasmid was modified to contain a separate EGFP or mEGFP expression cassette. Then, the pHAD plasmid was linearized by $PmeI$ enzyme to expose the ends of the 5' and 3' inverted terminal repeats (ITRs) and transfected into 116 producer cells (Profection Mammalian Transfection System, Promega, Madison, WI, USA). Helper virus (HV) was added the following day for HdAd production. Forty-eight hours postinfection, after cytopathic effects have taken place, cells were subjected to three freeze/thaw cycles for lysis and release of the viral particles. HdAd was purified by CsCl ultracentrifugation and stored at –80 °C in storage buffer (10 mM HEPES, 1 mM MgCl$_2$, 250 mM sucrose, pH 7.4).

## Virus injections

Virus injections at P14 were performed as previously described ($Lübbert\ et\ al.,\ 2019$). Briefly, mice were anesthetized with 5% isoflurane inhalation and anesthesia was maintained with 2% isoflurane throughout the procedure. Subcutaneous injection of physiological saline, lidocaine, bupivacaine, and meloxicam was used to treat loss of liquid and alleviate pain. The injection site was determined using the Stereodrive software (Neurostar) and corrected for head orientation and tilt. A small hole (<1 mm diameter) was drilled into the skull using a foot-pedal controlled drill (MH-170, Foredom). The virus solution was injected via a glass pipet (Drummond) at a rate of 100 nL/min with a nanoliter injector (NanoW, Neurostar). Following the injection, the glass needle was left in place for 1 min to dissipate the pressure and then slowly retracted. Animals were then placed under an infrared heat lamp and allowed to recover before being returned to their respective cages with their mother.

## Confocal imaging and reconstruction of presynaptic terminals

For reconstruction of the calyx of Held terminals, animals were injected with viral vectors expressing either mEGFP (yielding $Rac1^{+/+}$) or Cre and mEGFP (yielding $Rac1^{-/-}$). Mice were anesthetized with an intraperitoneal injection of tribromoethanol (250 mg/kg body weight) and transcardially perfused with ice-cold 0.1 M Sørensen's phosphate buffer (PB, pH 7.4) followed by 4% paraformaldehyde (PFA) in 0.1 M PB. Brains were removed and post-fixed overnight in 4% PFA solution at 4 °C. The next day, brains were sliced into 40 µm sections on a Leica VT1200 vibratome and mEGFP positive slices were mounted on cover slips with Aqua Polymount (Polysciences, Inc, Warrington, PA, USA). Confocal images were acquired with a Zeiss LSM 700 or 880 confocal scanning microscope using a 63 x/1.3 NA apochromat multi-immersion objective. Image stacks were collected using 0.44 µm plane line scans with line average of four times. Images were processed using Fiji ($Schindelin\ et\ al.,\ 2012$; http://fiji.sc; RRID:SCR_002285). Calyx reconstructions were performed blind to genotype using Imaris Measurement Pro (BitPlane) with automatic signal detection followed by manual curation in single planes of the z-stack confocal images as previously described ($Radulovic\ et\ al.,\ 2020$).

## Transmission electron microscopy (TEM)

Preembedding immuno-electron microscopy was performed as previously described (*Montesinos et al., 2015*; *Dong et al., 2018*). Briefly, *Rac^{fl/fl}* mice injected at P14 with HdAd co-expressing Cre and EGFP were anesthetized and perfused transcardially at P28 with phosphate-buffered saline (PBS, 150 mM NaCl, 25 mM PB, pH 7.4) followed by fixative solution for 7–9 min containing 4% PFA, 0.5% glutaraldehyde, and 0.2% picric acid in 100 mM PB (pH 7.4). Brains were post-fixed with 4% PFA in PB overnight and 50 µm coronal sections of the brainstem were obtained on a vibratome (Leica VT1200S). Expression of EGFP at the calyx of Held terminals was visualized using an epifluorescence inverted microscope (CKX41, Olympus) equipped with an XCite Series 120Q lamp (Excelitas Technologies), and only those samples showing EGFP were processed further. After washing with PB, sections were cryoprotected with 10%, 20%, and 30% sucrose in PB, submersed into liquid nitrogen and then thawed. Sections were incubated in a blocking solution containing 10% normal goat serum (NGS) and 1% fish skin gelatin (FSG) in 50 mM Tris-buffered saline (TBS, 150 mM NaCl, 50 mM Tris, pH 7.4) for 1 hr, and incubated with an anti-GFP antibody (0.1 µg/mL, ab6556, Abcam, RRID:AB_305564) diluted in TBS containing 1% NGS, 0.1% FSG, and 0.05% NaN$_3$ at 4°C for 48 hr. After washing with TBS, sections were incubated overnight in nanogold conjugated donkey anti-rabbit IgG (1:100, Jackson Immunoresearch, RRID:AB_2340609) diluted in TBS containing 1% NGS and 0.1% FSG. Immunogold-labeled sections were washed in PBS, briefly fixed with 1% glutaraldehyde in PBS, and silver intensified using an HQ silver intensification kit (Nanoprobe). After washing with PB, sections were treated with 0.5% OsO$_4$ in 0.1 M PB for 20 min, en-bloc stained with 1% uranyl acetate, dehydrated and flat embedded in Durcupan resin (Sigma-Aldrich). After trimming out the MNTB region, ultrathin sections were prepared at 40 nm thickness using an ultramicrotome (EM UC7, Leica). Sections were counterstained with uranyl acetate and lead citrate and examined in a Tecnai G2 Spirit BioTwin transmission electron microscope (Thermo Fisher Scientific) at 100 kV acceleration voltage. Images were taken with a Veleta CCD camera (Olympus) operated by TIA software (Thermo Fisher Scientific). Images used for quantification were taken at 60,000x magnification.

## TEM image analysis

Calyces positive for Cre expression (*Rac1^{−/−}*) were identified by immunogold labeling with an anti-GFP antibody and compared to EGFP-negative terminals (*Rac1^{+/+}*) in the same slice. All TEM data were analyzed using Fiji imaging analysis software (*Schindelin et al., 2012*). Each presynaptic active zone (AZ) was defined as the membrane directly opposing the postsynaptic density, and the length of each AZ was measured. Vesicles within 200 nm from each AZ were manually selected and their distances relative to the AZ were calculated using a 32-bit Euclidean distance map generated from the AZ. Synaptic vesicle distances were binned every 5 nm and counted (*Montesinos et al., 2015*; *Dong et al., 2018*). Vesicles less than 5 nm from the AZ were considered "docked". Three animals for each condition and ~40 AZs per animal were analyzed. Three researchers performed the analysis independently and blind to genotype and results were averaged.

## Electrophysiology

Acute coronal brainstem slices (~200 µm) containing the MNTB were prepared as previously described (*Chen et al., 2013*; *Thomas et al., 2019*). Briefly, after decapitation of the animal, the brain was immersed in low-calcium artificial cerebrospinal fluid (aCSF) solution containing (in mM): 125 NaCl, 2.5 KCl, 3 MgCl$_2$, 0.1 CaCl$_2$, 10 glucose, 25 NaHCO$_3$, 1.25 NaH$_2$PO$_4$, 0.4 L-ascorbic acid, 3 myo-inositol, and 2 Na-pyruvate (pH 7.3–7.4). Brain slices were obtained using a Leica VT 1200 S vibratome equipped with zirconia ceramic blades (EF-INZ10, Cadence Blades). The blade was advanced at a speed of 20–50 µm/s. Slices were immediately transferred to an incubation beaker containing recording aCSF (same as above but using 1 mM MgCl$_2$ and 1.2 mM CaCl$_2$) at ~37 °C and continuously bubbled with 95% O$_2$–5% CO$_2$. After approximately 45 min of incubation, slices were transferred to a recording chamber with the same aCSF at physiological temperature (~37 °C).

Electrical stimulation of afferent fibers was performed as previously described (*Forsythe and Barnes-Davies, 1993*). Briefly, a bipolar platinum-iridium electrode (FHC, Model MX214EP) was positioned medially of the MNTB to stimulate afferent fibers. Postsynaptic MNTB neurons were whole-cell voltage-clamped at −60 mV using an EPC10/2 amplifier controlled by Patchmaster Software (version 2x90.2, HEKA Elektronik, RRID:SCR_000034). Slices were continuously perfused with standard aCSF

solution at a rate of 1 mL/min and visualized by an upright microscope (BX51WI, Olympus) through a 60x water-immersion objective (LUMPlanFL N, Olympus) and an EMCCD camera (Andor Luca S, Oxford Instruments). To identify calyces expressing Cre and EGFP, the slice was illuminated at an excitation wavelength of 480 nm using a Polychrome V xenon bulb monochromator (TILL Photonics). For whole-cell voltage-clamp recordings, the standard extracellular solution was supplemented with 1 mM kynurenic acid to avoid saturation of postsynaptic AMPA receptors, 50 µM D-AP5 to block NMDA receptors, and 20 µM bicuculline and 5 µM strychnine to block inhibitory GABA and glycine receptors, respectively (all Tocris Bioscience). Patch pipettes had a resistance of 3–4 MΩ and were filled with the following (in mM): 130 Cs-gluconate, 20 tetraethylammonium (TEA)-Cl, 10 HEPES, 5 Na$_2$-phosphocreatine, 4 Mg-ATP, 0.3 Na-GTP, 6 QX-314, and 5 EGTA (pH 7.2, 315 mOsm). Reported voltages are uncorrected for a calculated liquid junction potential of 13 mV. For loose-patch recordings, the extracellular solution was supplemented with 20 µM bicuculline and 5 µM strychnine, and the patch pipettes were filled with aCSF. To simulate in vivo activity levels, afferent fibers were stimulated with activity patterns previously recorded in vivo in response to sinusoidal amplitude-modulated (SAM) sound stimulation (*Tolnai et al., 2008*).

mEPSCs were recorded with the same aCSF supplemented with 50 mM D-AP5, 20 µM bicuculline, 5 µM strychnine, 1 µM TTX, and 20 mM TEA. In a subset of *Rac1$^{-/-}$* recordings, Cd$^{2+}$ (cadmium chloride), a non-selective Ca$^{2+}$ channel blocker was flushed in during recordings to determine the impact of VGCC on spontaneous SV release. The baseline mEPSC frequency was established by recording each cell for at least one minute before Cd$^{2+}$ was flushed in via the bath perfusion. MNTB principal neurons were whole-cell voltage-clamped at –80 mV and recorded until enough mEPSCs (>25) were recorded. Overlapping events were excluded from the analysis.

Data were acquired at a sampling rate of 100 kHz and lowpass filtered at 6 kHz. Series resistance (3–8 MΩ) was compensated online to <3 MΩ, except for mEPSC recordings where series resistance (<9 MΩ) was not compensated. All experiments were performed at near-physiological temperature (36–37°C), and the temperature was maintained by a heated bath chamber (HCS, ALA Scientific Instruments) and perfusion system (HPC-2, ALA Scientific Instruments). The temperature of the bath solution was monitored during the experiment using a glass-coated micro thermistor.

## Electrophysiological data analysis

Electrophysiological data were imported to Matlab (version 9.10; The Mathworks, RRID:SCR_001622) using a custom-modified version of sigTool (*Lidierth, 2009*) and Igor Pro (version 8.0.4.2, Wavemetrics, RRID:SCR_000325) equipped with Patcher's Power Tools (version 2.19, RRID:SCR_001950) and NeuroMatic (*Rothman and Silver, 2018*) (RRID:SCR_004186), and analyzed offline with custom-written functions in Matlab and IgorPro. The remaining series resistance was compensated offline to 0 MΩ with a time lag of 10 µs (*Traynelis, 1998*). EPSC amplitudes were measured as peak amplitude minus baseline preceding the EPSC. RRP size and $P_r$ were calculated using the EQ method (*Elmqvist and Quastel, 1965*), back-extrapolation method (SMN with correction) (*Neher, 2015*), and NpRf model (*Thanawala and Regehr, 2016*) as previously described (*Lübbert et al., 2019*). Onset time of EPSCs was determined by fitting a Boltzmann function to the EPSC rising flank and calculating the time point of maximum curvature, as described previously (*Fedchyshyn and Wang, 2007*). The duration between stimulus and EPSC onset was defined as EPSC onset delay. The synchronicity of SV release for individual EPSCs was estimated from the 'effective EPSC duration' by dividing EPSC charge by EPSC amplitude. This 'effective EPSC duration' indicates the width of a square current pulse with amplitude and charge identical to that of the EPSC (*López-Murcia et al., 2019*). A shorter 'effective EPSC duration' indicates a shorter release transient and more synchroneous release provided that mEPSC kinetics are unchanged.

Recordings of mEPSCs were analyzed using NeuroMatic in Igor Pro. Potential events were detected when exceeding an amplitude threshold at 4–6 times the standard deviation of the baseline and all events were manually curated to exclude false positives. Rise time was measured between 10% and 90% of the peak amplitude.

Recovery of SVs was estimated by non-linear least-square fits to single EPSC (EPSC$_{test}$) and RRP (SMN method) recovery time courses. Both types of recovery time courses were best fit with a bi-exponential function of the form:

$$A \left( 1 - \left( f \left( e^{\frac{-t}{\tau_1}} \right) + (1-f) \left( e^{\frac{-t}{\tau_2}} \right) \right) \right)$$

and an F-test was used to determine the better fit. Fractional recovery of $EPSC_{test}$ was calculated as $\left( EPSC_{test}^{1} - EPSC_{cond}^{ss} \right) / \left( EPSC_{cond}^{1} - EPSC_{cond}^{ss} \right)$ where $EPSC_{test}^{1}$ is the amplitude of the first EPSC of the test train, $EPSC_{cond}^{ss}$ the steady-state amplitude of the conditioning train and $EPSC_{cond}^{1}$ the amplitude of the first EPSC of the conditioning train.

To estimate preceding neuronal activity during SAM stimulation, we calculated the interspike interval (ISI) as the distance to the preceding postsynaptic AP. To analyze AP responses during SAM stimulation, we used the preceding ISI to estimate the influence of previous activity during SAM recordings. Since short-term plasticity can extend well beyond the last ISI, we also estimated preceding activity by calculating the sum of all preceding events, with each event weighted by its distance to the AP under observation. This weighting was implemented as a single-exponential decaying function, emphasizing temporally close events over more distant ones (*Sonntag et al., 2011*; *Keine et al., 2016*). The time constant was set to 30 ms, consistent with previous reports (*Yang and Xu-Friedman, 2015*). For comparison, preceding activity levels were also estimated with time constants of 10 ms and 100 ms and yielded similar results.

## Experimental design and statistical analysis

Individual neurons were considered independent samples for electrophysiological data analysis and morphological reconstruction. For thin-section TEM analysis, AZs were considered independent samples (244 total AZs, 6 animals, ~40 AZs each). Statistical analysis was conducted in MATLAB and GraphPad Prism (RRID:SCR_002798). Data distributions were tested for Gaussianity using the Shapiro-Wilk test. To compare two groups, we used a two-tailed unpaired Student's t-test with Welch's correction (normal distribution) or a two-tailed Mann-Whitney $U$ test (non-normal distribution). An RM ANOVA was performed to compare more than two groups with within-subject factors, and p-values were Bonferroni-adjusted for multiple comparisons. Fits to data were subject to F-tests to determine the better model (mono- or bi-exponential) and for comparison between groups. A p-value of 0.05 was deemed significant for interpreting all statistical tests. Effect sizes were calculated using the MES toolbox in MATLAB (*Hentschke and Stüttgen, 2011*) and are reported as Cohen's $U_1$ for two-sample comparison and eta-squared ($\eta^2$) for RM ANOVA. No statistical test was performed to pre-determine sample sizes. Exact p-values, test statistics, and effect sizes for all statistical comparisons are summarized in the supplementary files. Boxplots show median, interquartile, and minimum/maximum within 1.5 times the interquartile range. Average data in the text are reported as mean ± standard deviation.

## Numerical simulations of STP and EPSC recovery

Simulations of synaptic STP in response to 50 and 500 Hz stimulus trains and of the recovery of EPSC amplitudes after SV pool depletion were performed using two types of kinetics schemes for SV priming and fusion: (*i*) a simple single-pool model with a $Ca^{2+}$-dependent SV pool replenishment as described below, and (*ii*) the sequential two-step model as recently proposed by *Lin et al., 2022*.

### Single-pool model

The simple single-pool model consisted of a single type of release site to which SVs can reversibly dock. The total number of sites ($N_{total}$) at any given time point $t$ is the sum of empty release sites ($N_e(t)$) and the release sites occupied by a primed and fusion-competent SV ($N_o(t)$): $N_{tot} = N_e(t) + N_o(t)$.

Transitions between $N_e(t)$ and $N_o(t)$ are described by forward ($k_f$) and backward ($k_b$) rate constants according to:

$$\frac{d}{dt} N_o(t) = k_f \cdot N_e(t) - k_b \cdot N_o(t)$$

$$\frac{d}{dt} N_e(t) = k_b \cdot N_o(t) - k_f \cdot N_e(t)$$

While the backward (unpriming) rate constant $k_b$ had a fixed value, $k_f$ was assumed to increase with cytosolic $[Ca^{2+}]$ ('effective $[Ca^{2+}]$'). The $Ca^{2+}$-dependence of $k_f$ was described by a Michaelis-Menten-like saturation according to:

$$k_f(t) = (k_{f,rest} + \sigma \cdot ([Ca^{2+}](t) - [Ca^{2+}]_{rest}))/(1 + ([Ca^{2+}](t) - [Ca^{2+}]_{rest})/K_{0.5})$$

where $k_{f,rest}$ is the value of $k_f$ at resting $[Ca^{2+}]$ ($[Ca^{2+}]_{rest}$; assumed to be 50 nM), $\sigma$ is a slope factor and $K_{0.5}$ is the Michaelis-Menten $K_D$ value.

The effective $[Ca^{2+}]$ ($[Ca^{2+}](t)$) was assumed to increase instantaneously at AP arrival and to decay back to its resting value $[Ca^{2+}]_{rest}$ with a rate constant $k_{Ca}$ according to the rate equation:

$$\frac{d}{dt}[Ca^{2+}](t) = -k_{Ca} \cdot ([Ca^{2+}](t) - [Ca^{2+}]_{rest})$$

Release probability ($P_r$) at arrival of the $j^{th}$ AP was modelled according to

$$P_{r,j} = P_{r,1} \cdot y_j^{4.5} \cdot z_j$$

with $y \geq 1$ and $z \leq 1$. Here, $P_{r,1}$ designates the release probability for the first EPSC in a train, $y_j$ accounts for changes in local $[Ca^{2+}]$ during repetitive stimulation ($y_j = [Ca^{2+}]_j/[Ca^{2+}]_1$), likely due to presynaptic $Ca^{2+}$ current facilitation, and/or saturation of local $Ca^{2+}$ buffers, and $z_j$ accounts for a small reduction of $P_r$ during repetitive stimulation.

Both variables $y_j$ and $z_j$ were initialized to 1 at the onset of a stimulus train. The variable $y$ was incremented after each AP by

$$y_{inc} = y_{inc,1} \cdot (y_{max} - y_j)$$

and $z$ was decremented by

$$z_{dec} = z_{dec,1} \cdot (z_j - z_{min})$$

During inter-stimulus intervals, time courses of $y(t)$ and $z(t)$ were determined by the rate equations:

$$\frac{d}{dt}y(t) = (1 - y(t)) \cdot k_y$$
$$\frac{d}{dt}z(t) = (1 - z(t)) \cdot k_z$$

(for details see **Lin et al., 2022**).

For each release event, the quantal content $m_j$ of the EPSC$_j$ triggered by stimulus $j$ was calculated as the product of $P_{r,j} \cdot N_o(t_j)$ with both quantities evaluated immediately before stimulus arrival. Between APs, the differential equations were solved numerically using the fifth-order Runge-Kutta-Fehlberg algorithm implemented in IgorPro. All model parameters on the time course of effective $[Ca^{2+}]$ and $P_r$ during stimulus trains were constrained to the same values for both genotypes. The remaining model parameters ($N_{total}$, initial $P_r$, $k_f$, $k_b$, $\sigma$, $K_{0.5}$) were adjusted by trial and error to reproduce experimentally observed differences between $Rac1^{+/+}$ and $Rac1^{-/-}$ synapses (**Figure 6—figure supplement 1**). The total number of SV docking sites ($N_{total}$), priming and unpriming rate constants, $Ca^{2+}$-dependence of the priming step, and $P_r$ were free parameters and adjusted by trial and error to reproduce experimental data on (1) the time course of fractional recovery after 500 Hz conditioning stimuli (**Figure 6A3**), (2) initial synaptic strength, and (3) the time course of STP during 50 and 500 Hz stimulation (**Figure 6A4**).

## Sequential two-step model

The sequential two-step model was implemented as previously described (**Lin et al., 2022**), with the standard model parameters modified to reproduce experimentally observed STP and EPSC recovery after conditioning trains in >P28 mouse calyx synapses and at physiological temperature. All model parameters defining the time course of effective $[Ca^{2+}]$ and $P_r$ during stimulus trains were constrained to the same values for both genotypes. Except for initial $P_r$ and $N_{total}$, the remaining model parameters were adjusted by trial and error to reproduce experimentally observed differences between $Rac1^{+/+}$ and $Rac1^{-/-}$ synapses. Thus, only those parameters determining the SV state transitions ($k_1$, $b_1$, $\sigma_1$, $K_{0.5}$,

$k_2$, $b_2$, $\sigma_2$) were allowed to differ between $Rac1^{+/+}$ and $Rac1^{-/-}$ synapses while initial $P_r$ and $N_{total}$ identical for both genotypes (*Figure 6—figure supplement 2*).

## Acknowledgements

We thank the members of the Young lab for their comments on the manuscript. We thank Danielle Yanda for mouse colony maintenance and preparation of tissue samples for electron microscopy experiments.

## Additional information

### Funding

| Funder | Grant reference number | Author |
|---|---|---|
| National Institute on Deafness and Other Communication Disorders | R01 DC014093 | Samuel M Young Jr |
| National Institute of Neurological Disorders and Stroke | R01 NS110742 | Samuel M Young Jr |
| Deutsche Forschungsgemeinschaft | 420075000 | Christian Keine |
| Max Planck Institute for Multidisciplinary Sciences | open access funding | Mrinalini Ranjan Holger Taschenberger |
| Max Planck Florida Institute for Neuroscience | open access funding | Connon I Thomas Debbie Guerrero-Given Naomi Kamasawa |
| University of Iowa | University Funds and University of Iowa Healthcare CCOM Distinguished Scholar Award | Samuel M Young Jr |

The funders had no role in study design, data collection and interpretation, or the decision to submit the work for publication.

### Author contributions

Christian Keine, Data curation, Software, Formal analysis, Validation, Investigation, Methodology, Writing – original draft, Writing – review and editing; Mohammed Al-Yaari, Connon I Thomas, Data curation, Formal analysis, Validation, Investigation, Writing – review and editing; Tamara Radulovic, Data curation, Formal analysis, Investigation, Writing – review and editing; Paula Valino Ramos, Data curation, Formal analysis; Debbie Guerrero-Given, Data curation, Formal analysis, Validation, Methodology, Writing – review and editing; Mrinalini Ranjan, Software, Visualization; Holger Taschenberger, Software, Visualization, Methodology, Writing – review and editing; Naomi Kamasawa, Data curation, Formal analysis, Validation, Investigation, Methodology, Writing – review and editing; Samuel M Young Jr, Conceptualization, Resources, Data curation, Formal analysis, Supervision, Funding acquisition, Validation, Methodology, Writing – original draft, Project administration, Writing – review and editing

### Author ORCIDs

Christian Keine  http://orcid.org/0000-0002-8953-2593
Mohammed Al-Yaari  http://orcid.org/0000-0003-3196-266X
Tamara Radulovic  http://orcid.org/0000-0002-2825-9773
Connon I Thomas  http://orcid.org/0000-0003-0995-9667
Mrinalini Ranjan  http://orcid.org/0000-0003-4310-3811
Holger Taschenberger  http://orcid.org/0000-0003-3186-3231
Naomi Kamasawa  http://orcid.org/0000-0002-8926-5309
Samuel M Young Jr,  http://orcid.org/0000-0002-7589-7612

## Ethics

All experiments were performed following animal welfare laws and approved by the Institutional Committee for Care and Use of Animals at the University of Iowa PHS Assurance No. D16-00009 (A3021-01) (Animal Protocol 0021952) and complied with accepted ethical best practices.

## Decision letter and Author response

Decision letter https://doi.org/10.7554/eLife.81505.sa1
Author response https://doi.org/10.7554/eLife.81505.sa2

---

# Additional files

## Supplementary files

• MDAR checklist

## Data availability

All numerical data used to generate the figures are part of the respective source files. Experimental raw data and custom-written software central to the conclusion of this study are available at https://doi.org/10.17632/c4b7gn8bh7 under the terms of the Creative Commons Attribution 4.0 License (CC BY 4.0).

The following dataset was generated:

| Author(s) | Year | Dataset title | Dataset URL | Database and Identifier |
|---|---|---|---|---|
| Keine C, Al-Yaari M, Radulovic T, Thomas CI, Valino Ramos P, Guerrero-Given D, Ranjan M, Taschenberger H, Kamasawa N, Young Jr SM | 2022 | Data/Software: Presynaptic Rac1 Controls Synaptic Strength Through the Regulation of Synaptic Vesicle Priming | https://doi.org/10.17632/c4b7gn8bh7 | Mendeley Data, 10.17632/c4b7gn8bh7 |

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
