## [Editor Report]

Keine et al. study the roles of the RhoGTPase Rac1 and actin in neurotransmitter release by ablating Rac1 at an age when synapses are essentially mature, thereby minimizing developmental compensations. They describe compelling findings supporting an increase in synaptic strength, interpreted as either an increase in release probability or priming of synaptic vesicles. Although direct support for Rac1-dependent altered presynaptic actin is not provided, the study delivers fundamental functional information on the role of Rac1 in regulating presynaptic release.

---

## [Decision Letter]

**Decision letter after peer review:**

Thank you for submitting your article "Presynaptic Rac1 controls synaptic strength through the regulation of synaptic vesicle priming" for consideration by *eLife*. Your article has been reviewed by 2 peer reviewers, and the evaluation has been overseen by a Reviewing Editor and John Huguenard as the Senior Editor. The following individual involved in the review of your submission has agreed to reveal their identity: Erwin Neher (Reviewer #2).

Essential revisions:

Please note in the detailed reviews below many questions about the clarity of approach and interpretation. In addition, two issues of higher impact are the following:

1) One of the most important issues is the clarification of the question about recovery controlling p_r, or vice versa – either by explaining what their argument is or else dropping that claim.

2) A more transparent discussion of the experiments regarding EPSC delays is required, as they may not directly contribute to the interpretation of results.

*Reviewer #2 (Recommendations for the authors):*

My major concern is the claim of release probability pr influencing the rate of recovery. Of course, there is a relationship between pr and rate of recovery at steady-state, since steady-state requires that recovery (SVs recovered per second) is equal to pr*frequency*steady-state pool size. But this does not imply that pr regulates recovery. Contrary to the statement that pr at steady-state can be derived from the data available, this reviewer holds (see comments for line 164) that calculation of pr,ss requires an independent evaluation of steady-state pool size.

Another major point is the ambiguity in the use of terms p_r, fusogenicity, and molecular versus positional priming. It seems that for most arguments the authors implicitly assume a single homogeneous pool. For others, heterogeneous fusogenicity has to be postulated. The question remains, whether this should be considered a difference in priming, in terms of 2-step priming or an actual difference in the fusion process.

In detail:

line 111: Statements, such as "This increase in synaptic strength is due to increased SV release probability, which results in faster SV replenishment and increased spontaneous SV release rates…" or "… regulating the intrinsic SV release probability through SV priming… " suggest a causal relationship between increased p_r and replenishment. But this has not been shown.

line 139: Quote Lee et al. (2012) for latrunculin.

line 154: Please specify, that you mean replenishment during trains and indicate, that you (probably) just calculate steady-state amplitudes*frequency. Thus, this evaluation does not convey new information.

line 150-159: Refer to individual panels in Figure 2.

line 158: 500 Hz steady-state is not affected – interesting! This excludes Rac1-control of an upstream limit to the priming process. Please point out, that you talk about replenishment during trains

line 164: pr,ss: EPSCss shorts out in this equation, which results in the equation for pr,1.

Your 'result' of constant p_r is a consequence of the assumption that the steady-state pool size scales with the amplitude ratio. This implies, that p_r is constant throughout the train.

line 165: This 'Slight increase in RRP', measured with SMN method, is almost a factor of 2 !!!. The claim, that there is no change in EQ and NpRf method is not substantiated (see below, comment on line 371). It is hard to believe, that the EQ method delivers the same pool estimates for the two traces in Figure 2, panel B2. There may be a problem in the selection of the region of the plot, to which the line fit is applied.

line 175: Once more: estimation of pr,ss assumes that the remaining RRP scales with EPSCss/ EPSCss.

line 195: wrong conclusion: you show that fusogenicity is independent of Ca-influx, but it may still be dependent on basal Ca.

line 225-229: Are decreased synchronicity and longer delay a consequence of larger release or of a deficit in positional priming or both… maybe with an additional effect of antagonizing Rab3. It may be interesting for the authors to have a look at Schlueter (2006) doi: 10.1523/JNEUROSCI.3553-05.2006.

Interaction between rab and rhoGTPases is quite common doi: 10.3390/cells8050396.

line 238-239: "Normalizing to the EPSC-amplitude … relative to … " is too complex. You actually show both simple normalization and normalized relative to steady (Figure 5 A2 and A3). The additional information in A3 is marginal. On the other hand, it seems that you are providing in the text only numbers regarding the slow component and nothing about relative amplitudes of components (or else: do you provide weighted time constants?)

line 312: You probably mean that steady-state RRP varies with stimulation frequency, which, however, is not the topic of the literature quoted here. Please note, that steady-state RRPsize is not measured in your study, and Lee et al. (2012) may be the only study, in which a serious attempt was made to do so.

line 324: Please point out, whenever you analyze replenishment whether you look at replenishment at rest (after depleting stimulation.) or else during train stimulation.

line 324-325: These results are not stated in the Results section.

line 326-341: Obscure, why and how pr controls the replenishment and why it does this on the time scale of hundred microseconds.

line 341-356: It seems that this argument builds on the assumption, that the steady-state pool is proportional to the ratio of steady-state EPSC over the first EPSC (see comment on line 164, above).

line 367-369: Kavalali would probably agree with the first half of this statement, but insists that synchronous and spontaneous releases originate from 2 populations of SVs and, therefore, does not support the conclusion of the second half.

line 371-373: EM resolution is not sufficient to reach this conclusion. RRP size, as measured here by AP train stimulation, does not allow the conclusion that RRP is not changed. One method does suggest a change; results from 2 other methods don't exclude a change. The statistical tests applied merely show that any differences (which are observed) cannot be declared as significantly different.

line 383-386: Yes, onset delay and decreased synchronicity point towards some deficit in positional priming, if not just an effect of larger release involving also more distant vesicles.

line 435: Are LS and TS states ever defined?

line 425 – 454: This reviewer's speculation is, that Rac1 exerts an antagonistic effect on Rab3, in addition to causing small defects in positional priming, unless the latter ones merely are a consequence of the increased release.

[Editors’ note: further revisions were suggested prior to acceptance, as described below.]

Thank you for resubmitting your work entitled "Presynaptic Rac1 controls synaptic strength through the regulation of synaptic vesicle priming" for further consideration by *eLife*. Your revised article has been evaluated by John Huguenard (Senior Editor) and a Reviewing Editor.

The manuscript has been improved but there are just a few remaining issues that need to be addressed, as outlined below, as outlined in the specific reviewer comments.

*Reviewer #1 (Recommendations for the authors):*

The addition of the numerical simulations was a good check on the conclusions, even if it resulted in a broad set of possibilities. As to the challenges of using EM to look for a change in actin, classical EM papers have described actin filaments associated with synapses, especially after using stabilization 'tricks' like phalloidin and tannic acid to keep the filaments intact upon processing. Whether they could work at the level you need, right at the membrane, is an experimental question for another time.

*Reviewer #2 (Recommendations for the authors):*

The authors have carefully considered all my points of criticism. They accepted that the size of the readily releasable pool of vesicles (RRP) cannot be determined during ongoing stimulation, which changed the interpretation of the data. Two interpretations are offered now: (1) in the framework of a single homogeneous RRP (2) assuming a 2-step priming process, followed by release, restricted to the fully-primed state. Adequate fits to the data can be obtained for both models if priming rate constants are assumed to be Ca-dependent. For the single pool model release probability, P_r, changes after ablation of Rac1, while with the 2-step model the required changes for describing the effects of ablation are restricted to priming steps. Comparing the 2 model predictions is interesting and may be helpful in the design of experiments to distinguish between these alternatives.

---

## [Author Response]

Essential revisions:Please note in the detailed reviews below many questions about the clarity of approach and interpretation. In addition, two issues of higher impact are the following:1) One of the most important issues is the clarification of the question about recovery controlling p_r, or vice versa – either by explaining what their argument is or else dropping that claim.2) A more transparent discussion of the experiments regarding EPSC delays is required, as they may not directly contribute to the interpretation of results.

We are very grateful to the reviewers for their careful analysis of our work. We thank both reviewers for their time, effort, and comments on the manuscript. We have extensively gone through the manuscript to address all points raised by the reviewers and especially the two issues of higher impact.

Regarding point (1):

In our manuscript, we present two main consequences following presynaptic deletion of Rac1: (1) an increase in synaptic strength accompanied by an increase in relative short-term depression, and (2) an acceleration of synaptic strength recovery after conditioning high-frequency AP trains. While the second finding directly relates to the SV pool replenishment kinetics and the priming step of the SV cycle, we also think that changes in *P_r_* can be linked to the SV priming process:

The result of the process of the molecular fusion apparatus assembly, which we refer to in our manuscript as ‘molecular priming’, defines the intrinsic fusogenicity of fusion-competent SVs. In addition, the spatial coupling between SVs and VGCCs which is referred to as ‘positional priming’ determines the local [ca^2+^] seen by the Ca sensor for SV fusion and thereby the probability with which a SVs fuses upon AP arrival. We tried to emphasize in the manscript that the result of these two aspects of SV priming – ‘molecular’ as well as ‘positional’ priming – can both influence *P_r_*.

In order to corroborate our conclusions about the underlying synaptic mechanisms of the observed changes, we now included numerical simulations of STP and EPSC recovery using two different STP models, a single-pool model and a sequential two-step model. These new results are presented in a new Figure 6 and related Figure Supplements.

Both STP models were capable of reproducing experimental findings related to Rac1 loss and both types of models required changes to model parameters determining the Ca-dependent priming kinetics. However, the conclusions regarding putative changes in *P_r_* were somewhat ambiguous. While the single-pool model required increased *P_r_* in *Rac1^-/-^* synapses to reproduce their increased synaptic strength, the sequential two-step model reproduced the observed differences by only changing the priming equilibrium which increased the abundance of tightly-docked SVs at rest. Both scenarios are now presented in the manuscript as possible mechanisms for Rac1 loss-induced changes in synaptic strength and STP.

Regarding point (2):

We observed increased synaptic strength after Rac1 loss, which may be due to increased *P_r_*. Shorter coupling distances between docked SVs and VGCCs represent one possible mechanism for increasing *P_r_*. This motivated us to evaluate EPSC delays and EPSC widths because we reasoned that shorter coupling distances will shorten the time it takes for entering ca^2+^ ions to diffuse and bind to the vesicular ca^2+^ sensor. At the same time, shorter SV to VGCC coupling distances may favor synchronous over asynchronous release by exposing SVs docked in the proximity of VGCCs to higher local [ca^2+^] and resulting in more fusion of SVs in the close vicinity of VGCCs.

We agree with this reviewer that additional clarifications were needed since we actually did not find experimental evidence supporting the hypothesis of a tighter coupling distance between SVs and VGCCs after Rac1 loss. We have therefore revised the respective sections in the Results and the Discussion.

Reviewer #2 (Recommendations for the authors):My major concern is the claim of release probability pr influencing the rate of recovery. Of course, there is a relationship between pr and rate of recovery at steady-state, since steady-state requires that recovery (SVs recovered per second) is equal to pr*frequency*steady-state pool size. But this does not imply that pr regulates recovery. Contrary to the statement that pr at steady-state can be derived from the data available, this reviewer holds (see comments for line 164) that calculation of pr,ss requires an independent evaluation of steady-state pool size.Another major point is the ambiguity in the use of terms p_r, fusogenicity, and molecular versus positional priming. It seems that for most arguments the authors implicitly assume a single homogeneous pool. For others, heterogeneous fusogenicity has to be postulated. The question remains, whether this should be considered a difference in priming, in terms of 2-step priming or an actual difference in the fusion process.

We thank the reviewer for pointing out the flaws and assumptions in our calculation of steady-state *P_r_*. We have removed our calculations of steady-state *P_r_* and the prior conclusions. To address the question of whether the enhanced synaptic strength can be explained by differences in priming, in terms of a two-step priming scheme, or by actual differences in the fusion process, we have now added new data based on numerical simulations using a single pool model or a two-step sequential priming model on our 50 Hz and 500 Hz train data and EPSC recovery data following 500 Hz conditioning train. This can be found as a new Figure 6 and text has been added to the Results and Discussion section.

Briefly, the numerical simulations used two STP models: (1) A single-pool model, that assumes a single type of release sites to which SVs can reversibly dock. Fusion-competent SVs constituting the RRP are homogenous with respect to *P_r_*, and (2) A two-step sequential model, which assumes SVs undergoing two sequential priming steps in order to assemble a mature release machinery. The model, distinguishes two SV priming states, a loosely-docked state (LS) with an immature fusion machinery, and a tightly-docked state (TS) with a mature fusion machinery, with only TS SVs being fusion competent. We then tested how well those two models can reproduce the 50 Hz and 500 Hz train data and EPSC recovery after 500 Hz conditioning trains. Both models were able to replicate the experimental data. To reproduce the observed differences, the single pool models required changes in three parameters, *N*_total_, SV priming kinetics and SV *P_r_*, while the two-step model required changes only in parameters determining SV priming kinetics. Even though changes in *N*_total_ or *P_r_* were not necessary in order to reproduce experimental findings using the sequential two-step scheme, we cannot rule out that such changes occur.

In conclusion, both STP models are in agreement that loss of Rac1 affects SV priming. Although the single-pool model supports changes in *P_r_*, which are in line with our analysis shown in Figure 2, the two-step model does not require such assumption. Our experimental data do not allow us to unambiguously favor one model over the other. We have revised and added text to discuss the interpretations based on the different assumptions underlying each STP model.

To address the reviewer’s concerns about the ”ambiguity in the use of terms p_r, fusogenicity, and molecular versus positional priming”, we carefully rewrote the corresponding text.

In detail:line 111: Statements, such as "This increase in synaptic strength is due to increased SV release probability, which results in faster SV replenishment and increased spontaneous SV release rates…" or "… regulating the intrinsic SV release probability through SV priming… " suggest a causal relationship between increased p_r and replenishment. But this has not been shown.

Based on our new Figure 6 and simulations we have now revised these statements in the text.

line 139: Quote Lee et al. (2012) for latrunculin.

Done.

line 154: Please specify, that you mean replenishment during trains and indicate, that you (probably) just calculate steady-state amplitudes*frequency. Thus, this evaluation does not convey new information.

We thank the reviewer for pointing this out. We have removed the data regarding replenishment during AP trains and report now only steady-state amplitudes and changed the corresponding text to clarify when either rates or rate constants were measured.

line 150-159: Refer to individual panels in Figure 2.

Done.

line 158: 500 Hz steady-state is not affected – interesting! This excludes Rac1-control of an upstream limit to the priming process. Please point out, that you talk about replenishment during trains

We agree that our data excludes Rac1 controlling of an upstream limit to the priming process. We explicitly state that we are discussing replenishment during trains.

line 164: pr,ss: EPSCss shorts out in this equation, which results in the equation for pr,1.Your 'result' of constant p_r is a consequence of the assumption that the steady-state pool size scales with the amplitude ratio. This implies, that p_r is constant throughout the train.

Thank you for pointing out this issue regarding the calculation of steady-state *P_r_*. We removed the calculation of steady-state *P_r_* and conclusions based on this calculation.

line 165: This 'Slight increase in RRP', measured with SMN method, is almost a factor of 2 !!!. The claim, that there is no change in EQ and NpRf method is not substantiated (see below, comment on line 371). It is hard to believe, that the EQ method delivers the same pool estimates for the two traces in Figure 2, panel B2. There may be a problem in the selection of the region of the plot, to which the line fit is applied.

We respectfully disagree that the RRP measured with the SMN method was increased by a factor of 2 in the *Rac1^-/-^* calyces. Using the SMN with correction we found that the increase in RRP between the *Rac1^+/+^* vs *Rac1^-/-^* is ~32% or a factor of 1.32 (means: 14.8 nA to 19.5 nA wt vs KO) or 43% (median).

We agree that the results of the EQ method depend on the selected fitting region. Therefore, we used several approaches to ensure that the data is fit correctly and consistently, despite the difference in short-term plasticity between *Rac1^+/+^* and *Rac1^-/-^*. We first fitted between the maximum EPSC and the following three EPSCs representing the area of steepest descent. The maximum EPSC in *Rac1*^−/−^ was typically the first EPSC, while in the *Rac1^+/+^* it was second or third EPSC. We found that results did not differ when fitting three or five consecutive EPSCs. We then repeated the analysis by calculating the steepest descend for each cell and fitting through this point plus the subsequent two data points. We found this approach to be the most reliable, and it does not change the results.

Specifically, calculation of the EQ plot for the average traces shown in Figure 2B2 estimated the average RRP at 14.9 nA for *Rac1^+/+^* and 16.9 nA for *Rac1^-/-^*, very similar to the 14.7 nA for *Rac1^+/+^* and 17.3 nA for *Rac1^-/-^* obtained from fitting single cells. Comparison of the EQ vs SMN plots are quite similar for *Rac1^+/+^* (14.8 nA vs 14.7 nA), while the EQ estimate in the *Rac1*^−/−^ is ~2 nA smaller than the SMN estimate (17.3 nA vs 19.5 nA). To help clarify our conclusions, we have added text to specifically point out that the RRP was larger and statistically different when using SMN correction compared to the other pool measurement methods.

**Author response image 1. sa2fig1:** Graphical representation of the EQ fit to the average data reported in Figure 2B2. The fitting area was determined by the steepest slope.

line 175: Once more: estimation of pr,ss assumes that the remaining RRP scales with EPSCss/ EPSCss.

See above, we have now removed the calculation of steady-state Pr from the manuscript and any respective conclusions.

line 195: wrong conclusion: you show that fusogenicity is independent of Ca-influx, but it may still be dependent on basal Ca.

We agree that we show fusogenicity is independent of Ca-influx and may still depend on basal Ca. Therefore, we have rephrased this sentence and added our premise based on our results from CAST/ELKS KO calyces which found an increase in mEPSCS in the presence of Cd^2+^ but no change in basal Ca levels. Please see lines 219-223 and lines 461-473.

line 225-229: Are decreased synchronicity and longer delay a consequence of larger release or of a deficit in positional priming or both… maybe with an additional effect of antagonizing Rab3. It may be interesting for the authors to have a look at Schlueter (2006) doi: 10.1523/JNEUROSCI.3553-05.2006.Interaction between rab and rhoGTPases is quite common doi: 10.3390/cells8050396.

We thank the reviewer for pointing us to the work by Schlüter et al. (2006).

The idea of Rac1 antagonizing Rab3 is interesting. While the loss of Rab3 impaired superpriming of SV, loss of Rac1 enhanced superpriming. However, the loss of Rac1 and loss of Rab3 have a similar phenotype with respect to recovery. Schlüter et al. (2006) show that in the absence of Rab3, EPSCs recover faster, similar to what is observed after the loss of Rac1. This suggests that Rac1 and Rab3s potentially regulate SV pool replenishment through similar mechanisms. One could argue that loss Rab3 results in a faster rate of SV priming due to increased SV mobility. Rac1 could lead to a restriction of SV mobility since it would promote more F-actin in the terminal. Based on the current literature in the field (Margiotta and Bucci Cells 2019. doi: 10.3390/cells8050396) there are multiple potential ways for Rac1 and Rabs to interact. However, future experiments will be needed to test these hypotheses.

line 238-239: "Normalizing to the EPSC-amplitude … relative to … " is too complex. You actually show both simple normalization and normalized relative to steady (Figure 5 A2 and A3). The additional information in A3 is marginal. On the other hand, it seems that you are providing in the text only numbers regarding the slow component and nothing about relative amplitudes of components (or else: do you provide weighted time constants?)

We apologize for the confusion. Figure 5A2 does not depict normalized data, but recovery of absolute EPSC amplitudes while Figure 5A3 shows normalized data (fractional recovery of EPSC_test_). The rationale is to show that both absolute and relative recovery is faster in *Rac1^−/−^.* Originally, to make the text more concise, we provide only weighted time constants in the text, but now we report in the text both slow and fast component as well as relative amplitudes.

line 312: You probably mean that steady-state RRP varies with stimulation frequency, which, however, is not the topic of the literature quoted here. Please note, that steady-state RRPsize is not measured in your study, and Lee et al. (2012) may be the only study, in which a serious attempt was made to do so.

We now removed the statement about steady-state RRP size.

line 324: Please point out, whenever you analyze replenishment whether you look at replenishment at rest (after depleting stimulation.) or else during train stimulation.

We have now added text to clarify that we are looking at replenishment at rest vs replenishment during ongoing train stimulation.

line 324-325: These results are not stated in the Results section.

Fitting results have now been added to the Results section.

line 326-341: Obscure, why and how pr controls the replenishment and why it does this on the time scale of hundred microseconds.

Based on our new simulation data, we have now rewritten this section.

line 341-356: It seems that this argument builds on the assumption, that the steady-state pool is proportional to the ratio of steady-state EPSC over the first EPSC (see comment on line 164, above).

The reviewer is correct that our argument was built on this assumption. As pointed out in prior responses to earlier critiques, we have now removed the steady-state Pr calculation and rewritten the respective discussion.

line 367-369: Kavalali would probably agree with the first half of this statement, but insists that synchronous and spontaneous releases originate from 2 populations of SVs and, therefore, does not support the conclusion of the second half.

We cannot comment if Kavalali would agree with the statement, however current data in the field supports both views, a single population of SVs that drives both synchronous and spontaneous release as well as the presence of two distinct populations of SVs responsible for either synchronous or spontaneous release. Regardless of having only a single or rather two SV populations, all SVs must undergo a priming step to acquire fusion competence. Actin is found throughout the presynaptic terminal and addition of latruculin increases mEPSC rates (Morales et al., 2000). Since our data supports that loss of Rac1 increases priming rates, increases in priming rates can lead to higher mEPSC rates by, for example, increasing the abundance of fusion competent TS SVs. We have rewritten the text to consider the origin of SVs for synchronous and spontaneous release. Please see lines 461-473 in the revised manuscript

line 371-373: EM resolution is not sufficient to reach this conclusion. RRP size, as measured here by AP train stimulation, does not allow the conclusion that RRP is not changed. One method does suggest a change; results from 2 other methods don't exclude a change. The statistical tests applied merely show that any differences (which are observed) cannot be declared as significantly different.

We agree that EM resolution may not be sufficient to make strong claims about RRP size changes. We used three common methods of RRP estimation (SMN, EQ and NpRf) and found that two out of three did not allow us to reject the null hypothesis of no difference in RRP size. All RRP estimate methods suggest that any potential increase in RRP after Rac1 loss must be less than 1.5 fold. This is too small to account for the 2-fold increase in synaptic strength. Therefore, and provided the assumptions of a single-pool model hold, *P_r_* has to increase significantly in Rac1-deficient synapses. We now rewrote the respective parts to avoid confusion.

Of note, our numerical simulations using the single-pool STP model also required a change in N_total_, the number of release sites, in addition to changes in SV priming rates and *P_r_* which would support a change in the RRP size as reported by the SMN method.

For comparison, we also applied an alternative correction for incomplete pool depletion to the SMN pool estimate by calculating the pool size for 50 Hz and 500 Hz stimulus train, plotting 1/RRP as a function of 1/f_stim_ and estimating a corrected RRP size at 1/f_stim_ = 0 by linear regression (Lipstein et al., 2021). We found those results to be similar to the SMN estimates reported in the manuscript (*Rac1^+/+^* = 14.1 ± 6.8 nA vs. *Rac1*^−/−^ = 21.6 ± 6 nA).

line 383-386: Yes, onset delay and decreased synchronicity point towards some deficit in positional priming, if not just an effect of larger release involving also more distant vesicles.

We agree that our data do not provide evidence for tighter coupling in the absence of Rac1.

line 435: Are LS and TS states ever defined?

The terms LS and TS is now defined in our presentation of the numerical simulations using the two-step sequential model.

line 425 – 454: This reviewer's speculation is, that Rac1 exerts an antagonistic effect on Rab3, in addition to causing small defects in positional priming, unless the latter ones merely are a consequence of the increased release.

Rac1 may exert an antagonistic effect on Rab3 as loss of Rab3 results in a loss of superprimed SVs, while loss of Rac1 enhances superpriming. However, both Rac1 and Rab3 lead to a similar phenotypes in regards to SV recovery which would indicate that they are not antagonistic. This is based on the EPSC recovery data from Figure 7A and 7C (Schlüter et al., 2006). The fractional recovery data in 7C demonstrated that the Rab3 ABCD KO recovered faster than the Rab BCD null. Based on these data, the authors conclude that in the absence of Rab3, EPSCs recover faster. In addition, the sucrose pool recovered faster in the absence of Rab3. Therefore, in regards to similar SV recovery phenotype, they would not be antagonistic.

As previously mentioned in our prior response about Rab3.

Loss of Rac1 and loss of Rab3 result in similar phenotypes in regards to SV recovery. Interestingly our Rac1 phenotype is similar to our GIT1/GIT2 KO phenotype described previously (Montesinos et al. 2015 Neuron) One could make the argument that GIT1/Rac1 and Rab3s are in a similar pathway that regulate SV release. One could hypothesize that Rab3 restricts SV mobility at the plasma membrane which results in a slower rate of SV priming. Rac1 would also restrict SV mobility since there would be an increase in F-actin. Another interesting observation in this context is that Piccolo (Waites et al 2011 J. Neuroscience) implicates F-actin regulation of SV release. An alternative hypothesis would be for Rab3 and Rac1 to directly interact and that loss of Rac1 results in an increase in Rac3 inactivated state which results in increased SV release.

Our data cannot rule out a change in positional priming, i.e. different SV to VGCC coupling, in Rac1-deficient synapses. But our EPSC onset delay data and also the data on effective EPSC durations do not provide evidence for tighter coupling. Numerical simulations based on release models that take into account the topology of docked SV with respect to the location of VGCCs may help to address these possibilities.

Lipstein N, Chang S, Lin KH, Lopez-Murcia FJ, Neher E, Taschenberger H, Brose N (2021) Munc13-1 is a Ca(2+)-phospholipid-dependent vesicle priming hub that shapes synaptic short-term plasticity and enables sustained neurotransmission. Neuron 109:3980-4000 e3987.

Morales M, Colicos MA, Goda Y (2000) Actin-dependent regulation of neurotransmitter release at central synapses. Neuron 27:539-550.

O'Neil SD, Racz B, Brown WE, Gao Y, Soderblom EJ, Yasuda R, Soderling SH (2021) Action potential-coupled Rho GTPase signaling drives presynaptic plasticity. *eLife* 10.

Schlüter OM, Basu J, Südhof TC, Rosenmund C (2006) Rab3 superprimes synaptic vesicles for release: implications for short-term synaptic plasticity. J Neurosci 26:1239-1246.

Taschenberger H, Scheuss V, Neher E (2005) Release kinetics, quantal parameters and their modulation during short-term depression at a developing synapse in the rat CNS. J Physiol 568:513-537.

Taschenberger H, Leao RM, Rowland KC, Spirou GA, von Gersdorff H (2002) Optimizing synaptic architecture and efficiency for high-frequency transmission. Neuron 36:1127-1143.

Yang CH, Ho WK, Lee SH (2021) Postnatal maturation of glutamate clearance and release kinetics at the rat and mouse calyx of Held synapses. Synapse 75:e22215.

[Editors’ note: further revisions were suggested prior to acceptance, as described below.]

Reviewer #1 (Recommendations for the authors):The addition of the numerical simulations was a good check on the conclusions, even if it resulted in a broad set of possibilities. As to the challenges of using EM to look for a change in actin, classical EM papers have described actin filaments associated with synapses, especially after using stabilization 'tricks' like phalloidin and tannic acid to keep the filaments intact upon processing. Whether they could work at the level you need, right at the membrane, is an experimental question for another time.

We thank the reviewer for their positive comments for pointing out that there are stabilization “tricks” that can keep actin filaments intact upon processing. We agree that testing these tricks is an experimental question for another time.

Reviewer #2 (Recommendations for the authors):The authors have carefully considered all my points of criticism. They accepted that the size of the readily releasable pool of vesicles (RRP) cannot be determined during ongoing stimulation, which changed the interpretation of the data. Two interpretations are offered now: (1) in the framework of a single homogeneous RRP (2) assuming a 2-step priming process, followed by release, restricted to the fully-primed state. Adequate fits to the data can be obtained for both models if priming rate constants are assumed to be Ca-dependent. For the single pool model release probability, P_r, changes after ablation of Rac1, while with the 2-step model the required changes for describing the effects of ablation are restricted to priming steps. Comparing the 2 model predictions is interesting and may be helpful in the design of experiments to distinguish between these alternatives.

We thank the reviewer for their positive comments. We agree that comparing the 2 model predictions is interesting and maybe helpful in designing future experiments to test these two alternatives.